# Enhanced Southern Ocean $CO_2$ outgassing as a result of stronger and poleward shifted southern hemispheric westerlies

Laurie C. Menviel[1,2*], Paul Spence[2,3], Andrew E. Kiss[4,5], Matthew A. Chamberlain[3,6], Hakase Hayashida[3,7], Matthew H. England[2,8], and Darryn Waugh[9,10]

[1]Climate Change Research Centre, University of New South Wales, Sydney, NSW 2052, Australia
[2]The Australian Centre for Excellence in Antarctic Science, University of New South Wales, Sydney, NSW 2052, Australia
[3]Institute for Marine and Antarctic Studies and Australian Antarctic Program Partnership, University of Tasmania, Hobart, Australia
[4]Research School of Earth Sciences, Australian National University, Canberra, Australia
[5]Australian Research Council Centre of Excellence for Climate Extremes, Australia
[6]CSIRO Oceans and Atmosphere, Hobart, Australia
[7]Application Laboratory, Japan Agency for Marine-Earth Science and Technology, Yokohama, Japan
[8]Centre for Marine Science and Innovation (CMSI), University of New South Wales, Sydney, NSW 2052, Australia
[9]School of Mathematics and Statistics, University of New South Wales, Sydney, NSW 2052, Australia
[10]Dpt. of Earth and Planetary Sciences, John Hopkins University, Baltimore, USA

**Correspondence:** L. Menviel (l.menviel@unsw.edu.au)

**Abstract.** While the Southern Ocean (SO) provides the largest oceanic sink of carbon, some observational studies have suggested that the SO total $CO_2$ ($tCO_2$) uptake exhibited large ($\sim$0.3 GtC/yr) decadal-scale variability over the last 30 years, with a similar SO $tCO_2$ uptake in 2016 as in the early 1990s. Here, using an eddy-rich ocean, sea-ice, carbon cycle model, with a nominal resolution of 0.1°, we explore the changes in total, natural and anthropogenic SO $CO_2$ fluxes over the period 1980-2021 and the processes leading to the $CO_2$ flux variability. The simulated $tCO_2$ flux exhibits decadal-scale variability with an amplitude of $\sim$0.1 GtC/yr globally in phase with observations. Notably, two stagnations in $tCO_2$ uptake are simulated: between 1982 and 2000, and between 2003 and 2011, while re-invigorations are simulated between 2000 and 2003, as well as since 2012. This decadal-scale variability is primarily due to changes in natural $CO_2$ ($nCO_2$) fluxes south of the polar front associated with variability in the Southern Annular Mode (SAM). Positive phases of the SAM, i.e. stronger and poleward shifted southern hemispheric (SH) westerlies, lead to enhanced SO $nCO_2$ outgassing due to higher surface natural dissolved inorganic carbon (DIC) brought about by a combination of Ekman-driven vertical advection and DIC diffusion at the base of the mixed layer. The pattern of the $CO_2$ flux anomalies indicate a dominant control of the interaction between the mean flow south of the polar front and the main topographic features. While positive phases of the SAM also lead to enhanced anthropogenic $CO_2$ ($aCO_2$) uptake south of the polar front, the amplitude of the changes in $aCO_2$ fluxes is only 25% of the changes in $nCO_2$ fluxes. Due to the larger $nCO_2$ outgassing compared to $aCO_2$ uptake as the SH westerlies strengthen and shift poleward, the SO $tCO_2$ uptake capability thus reduced since 1980 in response to the shift towards positive phases of the SAM. Our results indicate that, even in an eddy-rich ocean model, a strengthening and/or poleward shift of the SH westerlies enhance $CO_2$ outgassing. The projected poleward strengthening of the SH westerlies over the coming century will thus reduce the capability of the SO to mitigate the increase in atmospheric $CO_2$.

## 1 Introduction

As a result of anthropogenic emissions of greenhouse gases, atmospheric $CO_2$ concentration ($CO_{2atm}$) increased from a natural level of 277 ppm in 1750 (Joos and Spahni, 2008) to 415 ppm in 2021 (Friedlingstein et al., 2022). The terrestrial biosphere and the ocean have however strongly mitigated the anthropogenic emissions of carbon, respectively absorbing $\sim$31% and 24% of the emissions (Le Quéré et al., 2018). The largest oceanic carbon sink is the Southern Ocean (SO), which has contributed $\sim$40% of the global oceanic $CO_2$ uptake in the 1990s (Sabine et al., 2004; Mikaloff-Fletcher et al., 2006). In the context of continued anthropogenic emissions of greenhouse gases, it is crucial to better understand the impact of climate change on atmosphere-ocean total (sum of anthropogenic and natural) $CO_2$ ($tCO_2$) fluxes.

There is evidence for large decadal variability in the total SO carbon uptake (LeQuéré et al., 2007; Matear and Lenton, 2008; Landschützer et al., 2015; Bushinsky et al., 2019; Gruber et al., 2019; Keppler and Landschützer, 2019; Gruber et al., 2023). Observational estimates, covering the period 1982-2011, suggest the total carbon uptake in SO was lower than expected in the 1990s, but increased significantly between 2002 and 2011 to reach a maximum of 1.2 GtC yr$^{-1}$ in 2011 (LeQuéré et al., 2007; Landschützer et al., 2015; Gruber et al., 2019). Between 2011 and 2016, observational estimates combining the Surface Ocean $CO_2$ ATlas (SOCAT) and Southern Ocean Carbon and Climate Observations and Modeling (SOCCOM) suggest that the SO $tCO_2$ uptake weakened by $\sim$0.4 GtC yr$^{-1}$ (Bushinsky et al., 2019; Gruber et al., 2019; Keppler and Landschützer, 2019), while observational estimates only including SOCAT suggest the $tCO_2$ uptake stabilised between 2011 and 2016, and increased since 2017 (Landschützer et al., 2020). There is significant uncertainty associated with these estimates due to the sparsity of the data, particularly in the 1990s (Ritter et al., 2017; Gregor et al., 2018), and little information prior to 1982. In addition, this data scarcity might lead to a 39% overestimation of the amplitude of SO decadal $tCO_2$ uptake variability (Gloege et al., 2021).

The SO circulation is mostly driven by SH westerly winds, which generate an equatorward Ekman transport and an associated upwelling of carbon-rich deep waters. Changes in the position and strength of the SH westerlies are linked to the dominant mode of atmospheric variability in the southern hemisphere, the SAM. Positive SAM phases, which are associated with poleward contraction and stronger than average westerly winds, have been observed between 1979 and 2000, particularly during austral summer and autumn (Thompson and Solomon, 2002; Fogt and Marshall, 2020). This intensification and poleward shift of the SH westerlies result from stratospheric ozone depletion and an increase in greenhouse gases (Arblaster and Meehl, 2006). However, due to stratospheric ozone recovery, a pause in the poleward intensification of the SH westerlies has been observed between 2000 and $\sim$2010 (Banerjee et al., 2020). The SAM trend is however positive since $\sim$2010. The continued increase in greenhouse gases is taking over the ozone impact and is suggested to result in a long-term positive SAM trend over the 21st century (Thompson et al., 2011).

Numerical studies have highlighted the role of SH westerlies in modulating the upwelling of DIC-rich deep water and thus the carbon exchange between the atmosphere and the ocean. Stronger SH westerlies enhance the SO upwelling, leading to an oceanic loss of carbon and thus an increase in $CO_{2atm}$ (Toggweiler, 1999; Lauderdale et al., 2013; Lovenduski et al., 2007, 2008; Munday et al., 2014; Lauderdale et al., 2017; Menviel et al., 2018). Changes in the strength and position of the SH westerlies associated with the SAM could thus significantly modulate interannual SO $CO_2$ fluxes (Resplandy et al., 2015). The

increase in surface DIC as a result of stronger SO upwelling could however be partly mitigated by enhanced export production

at the surface of the SO (Menviel et al., 2008; Hauck et al., 2013). In addition, the impact of latitudinal changes in the position of the SH westerlies on oceanic carbon and $CO_{2atm}$ is uncertain as it might depend on the initial position of the SH westerlies and on how the latitudinal changes in the SH westerlies impact the oceanic circulation (Völker and Köhler, 2013; Lauderdale et al., 2013, 2017).

Most of the numerical studies analysing the impact of SH westerly changes mentioned above focused on natural carbon.

Given the increase in anthropogenic carbon emissions since 1870, the natural carbon cycle has been perturbed, and the impact of changes in the strength and position of the SH westerly winds on anthropogenic carbon uptake also needs to be taken into account. Only a few studies performed with coarse resolution ocean models (Lovenduski et al., 2007, 2008) have assessed the impact of the SAM on total, anthropogenic and natural $CO_2$ fluxes. They found that positive phases of the SAM led to an outgassing of natural $CO_2$ ($nCO_2$), while enhancing the uptake of $aCO_2$, with the net effect being a reduction in the $tCO_2$

uptake. Lenton and Matear (2007) also simulated a reduction in $tCO_2$ uptake in response to the SAM but did not distinguish between the natural and anthropogenic contributions.

Some studies have thus attributed the weaker SO carbon uptake observed in the 1990s to a positive trend in the SAM (Marshall, 2003; LeQuéré et al., 2007; Lenton and Matear, 2007; Lovenduski et al., 2007, 2008; Gruber et al., 2023). This is supported by recent observations-based studies, which concluded that the multi-decadal surface SO $pCO_2$ variability, particu-

larly the winter trend, was driven by the SAM (Gregor et al., 2018; Nevison et al., 2020), even though it was also suggested that on top of this multi-decadal trend net primary production could affect interannual variability (Gregor et al., 2018). On the other hand, McKinley et al. (2020) suggested that the lower 1990s $CO_2$ uptake was a response to the slower $CO_{2atm}$ growth rate.

Regarding, the re-invigoration of the SO carbon uptake in the 2000s, Landschützer et al. (2015) suggested that it could

not be attributed to the SAM because the ERA-interim reanalysis did not display the associated wind changes. Instead, they attributed the enhanced carbon uptake to increased solubility in the Pacific sector of the SO due to surface cooling, and a weaker upwelling of DIC-rich waters in the Atlantic and Indian sectors of the SO. More recently, by analysing changes in SO $tCO_2$ fluxes between 1980 and 2016, Keppler and Landschützer (2019) suggested that the net effect of the SAM on $tCO_2$ uptake was nil and that instead the variability was arising from regional shifts in SO surface air pressure linked to zonal wavenumber 3.

There are thus uncertainties not only in the magnitude of the decadal variability but also on the processes controlling SO carbon uptake, and there is a need for further studies examining these issues. In addition, the impacts of mesoscale eddy activity on the SO oceanic circulation and transport of nutrient and carbon needs to be better constrained. The prevalent mesoscale eddy activity in the SO significantly influences heat, salt and nutrient transport as well as the SO lateral and meridional overturning circulations. Mesoscale eddy transports generally act in the opposite sense to the wind driven transport in the SO, and thus the

response of the ocean circulation to changes in the winds varies across model studies. For example, a doubling of the magnitude of SH westerly winds doubles the simulated circumpolar transport in coarse resolution models that poorly parameterize eddies, but this doubling does not occur in eddy-resolving simulations (Hallberg and Gnanadesikan, 2006; Farneti et al., 2010; Spence et al., 2010; Dufour et al., 2012; Morrison and Hogg, 2013; Munday et al., 2013). Further, Dufour et al. (2013) showed

in an eddy-permitting ($\sim$0.5$°$ resolution) model that even though a strengthening and poleward shift of the SH westerlies, representing positive phases of the SAM, leads to stronger Ekman-induced northward natural DIC transport, a third of this is compensated by enhanced southward natural DIC transport through eddies. Dufour et al. (2013) also suggested that the higher surface DIC during positive phases of the SAM resulted from enhanced vertical diffusion at the base of the mixed layer and not from vertical advection.

Here, we analyze a simulation of the period 1980-2021 performed with an eddy-rich ocean, sea ice, biogeochemical model and forced with the 55-year Japanese Reanalysis for driving oceans (JRA55-do) (Tsujino et al., 2018) atmospheric fields to better understand the interannual to multi-decadal variability in SO natural, anthropogenic and total $CO_2$ fluxes and their links to changes in the SAM.

## 2 Methods

### 2.1 Models

Changes in SO carbon uptake are examined using a simulation performed with the eddy-rich ACCESS-OM2-01 global model configuration run under interannual forcing between 1958 and 2021. The ocean model is the Modular Ocean Model (MOM) version 5.1 (Griffies, 2012) with a nominal resolution of 0.1$°$ and 75 vertical levels increasing smoothly in thickness from 1.1 m at the surface to 198.4 m at the bottom (5808.7 m depth). The ocean model is coupled to the thermodynamic-dynamic Los Alamos sea-ice model (CICE) version 5.1.2 (Hunke et al., 2017). ACCESS-OM2 is described in detail in Kiss et al. (2020), but the version presented here has many improvements as described in Solodoch et al. (2022). The main improvements are that the wind stress calculation now uses relative velocity over both ocean and sea ice (not just ocean), and the albedo of the ocean is now latitude-dependent following Large and Yeager (2009).

The ocean biogeochemical model is the Nutrient-Phytoplankton-Zooplankton-Detritus (NPZD) model WOMBAT (Whole Ocean Model of Biogeochemistry And Trophic-dynamics) (Kidston et al., 2011; Oke et al., 2013; Law et al., 2017). WOMBAT includes DIC, CaCO$_3$, alkalinity, oxygen, phosphate and iron, that are linked to the phosphate uptake and remineralisation through a constant Redfield ratio. The biogeochemical parameters are identical to those of the ACCESS-ESM1.5 model (Ziehn et al., 2020), apart from the pre-industrial $CO_{2atm}$ value. Two DIC tracers are included, a natural DIC (nDIC) and a total DIC (tDIC), with the difference between the two providing an estimate of anthropogenic DIC (aDIC). nDIC exchanges carbon with a constant pre-industrial $CO_{2atm}$ concentration of 284.32 ppm, whereas tDIC exchanges carbon with the time evolving, observed $CO_{2atm}$, which includes the current increase due to anthropogenic emissions. For the tDIC tracer, the $CO_{2atm}$ concentration is spatially uniform and annually-averaged, and follows the Ocean Model Intercomparison Project (OMIP) protocol until 2014 (Orr et al., 2017) and NOAA GML data thereafter, rising from 315.34 ppm in 1958 to 339.7 ppm in 1980 and 414.72 ppm in 2021 (Fig. 2a). The air-sea $CO_2$ exchange is a function of the difference in partial pressure of $CO_2$ at the air-sea interface, the wind speed (Wanninkhof, 1992) and sea ice concentration. This version also includes a two-way coupling of the ocean biogeochemistry with nutrient and algae carried in the sea ice model (Hayashida et al., 2021). Note, physical and biogeochemical changes in the ocean simulations do not impact the atmospheric state, and in all simulations the longwave

radiative forcing is given by the evolving JRA55-do fields over 1970-2021. Biogeochemical tracers also have no effect on the ocean or sea ice physical state (including shortwave penetration depth), and oxygen has no effect on other biogeochemical tracers.

## 2.2 Simulations

The model is forced by atmospheric conditions taken from the 55-year Japanese Reanalysis for driving oceans (JRA55-do) version 1.4, 1.5.0 and 1.5.0.1 (Tsujino et al., 2018), that now covers the period 1958 to 2021. JRA55-do allows calculation of air-sea fluxes of momentum, heat and freshwater at a 3 hourly time interval and with a horizontal resolution of $0.5625°$. The model is forced by four 61-year cycles of JRA55-do v1.4 for the period 1958-2018. In cycle 1, the ocean model was initialised with modern-day temperature and salinity distributions derived from the World Ocean Atlas 2013 version 2 (Locarnini et al., 2013). Subsequent cycles used the final state of the previous cycle as the initial condition, and proceeded through another JRA55-do v1.4 1958-2018 forcing cycle. Cycles 1–3 do not include biogeochemistry.

Here we analyse cycle 4, which includes WOMBAT BGC and has also been extended from 2019 until the end of 2021 using JRA55-do v1.5.0 for 2019 and JRA55-do v1.5.0.1 thereafter. Ocean and ice physical fields were initialised from the values at the end of cycle 3. Biogeochemical fields other than oxygen were initialised at the start of cycle 4 (1958). A uniform $0.01$ mmol m$^{-3}$ initial value was used for phytoplankton, zooplankton, detritus and CaCO$_3$. Initial alkalinity, tDIC and nDIC were interpolated from a 305 years-long spinup run at 1 degree resolution (i.e., the first 5 cycles of OMIP2) (Mackallah et al., 2022). GLODAPv2 (Olsen et al., 2016) was used to initialise phosphate, and iron was initialised from the FEMIP median value (Tagliabue et al., 2016). Oxygen was initialised from GLODAPv2 at 1 Jan 1979 due to a configuration mistake; this has no effect on other variables. Initial phosphate and algae in the bottom 3 cm of sea ice were set to zero, but quickly equilibrate with those in the surface ocean layer. The physical state of the ACCESS-OM2-01 global model is consistently simulated across all four, 61-year forcing cycles. Here, we skip the first 22 years of the fourth cycle (i.e. 1958-1980) from our analysis to allow the simulation to recover from the reset at the end of the previous cycle, and focus our analysis on the period 1980-2021, following phase II of the Coordinated Ocean-ice Reference Experiments (CORE).

To better assess the impact of high model resolution, we also provide a comparison with the results of a $1°$ resolution configuration of ACCESS-OM2. This is based on that described by Kiss et al. (2020), but is forced with the newer JRA55-do v1.4 1958–2018 dataset and includes WOMBAT biogeochemistry. Specifically, this simulation corresponds to the final ($6^{th}$) cycle of omip2-spunup (Mackallah et al., 2022) after 33 cycles were performed as spin-up. Similar to ACCESS-OM2-01, in the version of ACCESS-OM2 used here the wind stress calculation uses relative velocity over both ocean and sea ice and the albedo of the ocean is latitude-dependent following Large and Yeager (2009). The $1°$ configuration has flow-dependent Gent and McWilliams (1990) parameterisation of unresolved mesoscale eddies, as described in Kiss et al. (2020).

## 2.3 Analysis

For consistency with the forcing, the SAM index is calculated from the JRA55-do dataset with the methodology described in Stewart et al. (2020). The SAM index as calculated from the JRA55-do dataset captures well the SAM index based on observations (Marshall, 2003; Stewart et al., 2020).

In this study the SO polar front (PF) and sub-Antarctic front (SAF) are defined as the 1.2°C annual minimum surface temperature contour, and 4°C isotherm at 400m depth, respectively following the definition of Sokolov and Rintoul (2009). Their simulated zonal mean latitude locations are 56.3°S and 48.2°S, respectively.

Oceanic natural $pCO_2$ is a function of nDIC, alkalinity (ALK) as well as ocean temperature (T) and salinity (Sal). Changes in $pCO_2$ can thus be described as:

$$\Delta pCO_2 = \frac{\partial(pCO_2)}{\partial(DIC)}.\Delta DIC + \frac{\partial(pCO_2)}{\partial(ALK)}.\Delta ALK + \frac{\partial(pCO_2)}{\partial(Sal)}.\Delta Sal + \frac{\partial(pCO_2)}{\partial(T)}.\Delta T \tag{1}$$

To better understand the processes leading to surface ocean $pCO_2$ changes, we can estimate the $pCO_2$ change from each of the above variables separately. Broecker et al. (1979) derived that if ALK, salinity and temperature are constant then:

$$\frac{\partial ln(pCO_2)}{\partial ln(DIC)} = \gamma DIC \tag{2}$$

with $\gamma DIC$ being the Revelle factor of DIC.

Equation 1 can be re-written as:

$$\frac{DIC}{pCO_2}.\frac{\partial pCO_2}{\partial DIC} = \gamma DIC \tag{3}$$

One can then derived the $pCO_2$ change due to a change in DIC ($\Delta pCO_{2DIC}$) as:

$$\Delta pCO_{2DIC} = \gamma_{DIC} pCO_2 \frac{\Delta DIC}{DIC} \tag{4}$$

Here we use a mean high latitude estimate for $\gamma_{DIC}$ of 13.3 (Sarmiento and Gruber, 2006) to estimate $pCO_{2DIC}$. $pCO_2$ sensitivities to ALK and salinity can be derived with similar equations:

$$\Delta pCO_{2ALK} = \gamma_{ALK} pCO_2 \frac{\Delta ALK}{ALK} \tag{5}$$

and

$$\Delta pCO_{2Sal} = \gamma_{Sal} pCO_2 \frac{\Delta Sal}{Sal} \tag{6}$$

with $\gamma_{ALK} = -12.6$ and $\gamma_{Sal} = 1$ (Sarmiento and Gruber, 2006).

Finally, Takahashi et al. (1993) suggest that the $pCO_2$ sensitivity to temperature (T) follows the relationship:

$$\frac{\partial ln(pCO_2)}{\partial T} = 0.0423°C^{-1} \tag{7}$$

This implies (Takahashi et al., 2002, 2009) that the change in $pCO_2$ due to temperature is:

$$\Delta pCO_{2T} = (e^{0.0423\Delta T} - 1)pCO_2 \qquad . \qquad (8)$$

Changes in oceanic remineralised carbon concentration ($Corg$) between 1980 and 2021 are estimated as follows:

$$\Delta Corg = R_{C/P}\Delta PO_{4Reg} \qquad (9)$$

with

$$\Delta PO_{4Reg} = R_{P/O2}\Delta AOU \qquad (10)$$

AOU is the apparent oxygen utilisation, and is the difference between the dissolved oxygen at saturation (as a function of temperature and salinity) and the simulated dissolved oxygen concentration. $R_{C/P}$ and $R_{P/O2}$ are the Redfield ratios equal to 106/1 and 1/172, respectively.

## 3  Results

### 3.1  Mean $CO_2$ flux patterns

Performance of the multi-resolution ACCESS-OM2 model suite is discussed in Kiss et al. (2020). The eddy-rich ACCESS-OM2-01 model at 0.1° resolution provides a reasonable and consistent representation of the state of the ocean, with a particularly good representation of the SO circulation structure, dense shelf water formation and abyssal overturning cell (Morrison et al., 2020). The simulated horizontal and vertical gradients in tDIC, alkalinity and dissolved oxygen ($O_2$) in the Southern Ocean are in good agreement with observations (Olsen et al., 2016) (Fig. S1). The absolute values of tDIC and alkalinity are $\sim$60 $\mu$mol/kg$^{-1}$ lower than those estimated by GLODAP. This bias might lead to a $\sim$ 10-15 ppm underestimation of the total $pCO_2$ of SO waters, which does not seem to significantly impact the SO $tCO_2$ uptake as detailed below. The concentrations of the different biogeochemical tracers are not constant through the simulation since the atmospheric forcing varies (among other reasons), however the trends averaged over the Southern Ocean and at different depths are much lower than 1% (Fig. S2). In addition, apart from the nDIC at intermediate depth and dissolved $O_2$ at depth, the trends are similar in the 0.1° and 1° resolution, even though the 1° resolution has been equilibrated for a longer time. This indicates that both the 0.1° and 1° simulations can be used to study the SO biogeochemical response to the atmospheric forcing.

We first assess the performance of the model by comparing the time-mean simulated SO $tCO_2$ fluxes to observational estimates (Fig. 1a,b). The SOCAT version 6 (Bakker et al., 2016) provides surface ocean $CO_2$ measurements. However, due to the spatial and temporal heterogeneity of these measurements, it does not provide an appropriate dataset for a comparison with simulated fields. To fill this gap, Landschützer et al. (2016) developed a method to provide a global gridded monthly observational estimate. The ocean is first clustered into biogeochemical provinces using Self-Organizing Map (SOM). Then, within each biogeochemical province, $pCO_2$ estimates are generated based on a non-linear relationship between the SOCATv6 observations

and the $CO_2$ driver variables through a feed-forward neural network (FFN) approach. If averaged over the available period of 1982-2021, the observationally-derived SOM-FFN dataset (Landschützer et al., 2020) (Fig. 1a) displays a strong $tCO_2$ uptake north of 50°S (-1.59 mol/m$^2$/yr, zonal average between 50°S and 35°S), and a weak $tCO_2$ uptake (-0.38 mol/m$^2$/yr) south of 50°S, even though there are some areas with outgassing (~0.2 mol/m$^2$/yr) south of 50°S.

These features are relatively well reproduced by the simulated $tCO_2$ fluxes (Fig. 1b), which display a similar strong uptake (-1.59 mol/m$^2$/yr) north of 50°S, that is generally north of the SAF (Sokolov and Rintoul, 2009). As in the observations, some $tCO_2$ outgassing (~0.4 mol/m$^2$/yr) is simulated south of the SAF, but particularly south of the PF. While both observational estimates and simulation suggest a $tCO_2$ outgassing south of the PF at $0 - 60$°E, $150$°E $- 180$°E, and downstream of the Drake passage, the simulated $tCO_2$ outgassing is particularly confined to some hotspots, namely over the eastern part of the Southeast Indian Ridge, east of Drake Passage and over the Southwest Indian Ridge (Fig. 1b). Overall, a similarly weak $tCO_2$ uptake (-0.59 mol/m$^2$/yr) is estimated south of 50°S.

These fluxes can be decomposed into their $nCO_2$ and $aCO_2$ components, thus highlighting an uptake of $aCO_2$ nearly everywhere south of 35°S (Fig. 1d), with two zonally-averaged maximum $aCO_2$ uptake at 42°S and 55°S (Fig. S3f). While this is broadly consistent with observational estimates, the observations suggest a maximum $aCO_2$ uptake at 50°S (Gruber et al., 2023). By contrast, an outgassing of $nCO_2$ is simulated south of the PF and in the frontal zone of the Indian sector (Figs. 1c and S3e).

The upwelling of DIC-rich deep waters south of the PF (Figs. 1e and S3c) and the subsequent northward advection of these waters (Fig. S3b) contribute to the $nCO_2$ outgassing south of 50°S. Through Ekman transport, surface waters in the SO move equatorward (Fig. S3b), and nutrients and DIC are consumed by phytoplankton, leading to a maximum detritus flux at ~42°S (Fig. S3d) and $nCO_2$ ocean uptake north of the SAF (Fig. S3e), where Antarctic Intermediate Waters (AAIW) and Subantarctic Mode Waters (SAMW) are formed. Within the model framework, the detritus flux at 100m depth provides an estimate of export production.

The pattern of mean simulated SO $CO_2$ fluxes are similar in the 0.1° (ACCESS-OM2-01) and 1° (ACCESS-OM2) version of the model (Figs. S3 and S4), implying a dominant effect of the large-scale oceanic circulation on the $CO_2$ fluxes. The main differences are that the $tCO_2$, $nCO_2$ and $aCO_2$ flux hotspots over the eastern part of the Southeast Indian Ridge, east of the Drake Passage and over the Southwest Indian Ridge are more pronounced in the ACCESS-OM2-01 (Fig. 1b,c,d) than in the ACCESS-OM2 (Fig. S4b,c,d), implying a stronger interaction between circulation and topography in the eddy-rich model.

## 3.2 Temporal changes in $CO_2$ fluxes

We now look at the time evolution of SO $CO_2$ fluxes since 1980 (Fig. 2). From 1980 to 2021, the SO $nCO_2$ uptake (Fig. 2c, grey and shading) reduces by ~0.28 GtC/yr (mean slope of 0.07 GtC/yr per decade). The $nCO_2$ uptake is stronger than the 1980-2021 mean before the mid-1990s and weaker after that (Fig. 2c). On top of the long-term trend, the $nCO_2$ flux displays a large (~0.15 GtC/yr) decadal-scale variability. $nCO_2$ fluxes are strongly correlated with the SAM index calculated from the JRA-55do dataset (R=0.62 for annual mean data and R=0.84 with a 5-year smoothing, Fig. 2b), with periods of weak $nCO_2$ uptake associated with positive phases of the SAM. This SAM link is largely related to changes in the strength of the zonal

SO wind stress (Fig. S5d, R=0.62 for yearly data, 0.92 with 5-year smoothing), even though a poleward displacement of the maximum wind stress also reduces the $nCO_2$ uptake (Fig. S5g, R=-0.4).

Since the SAM index displays a trend towards the positive phase between 1980 and 2021, the correlation mentioned above includes both interannual variability as well as decadal-scale changes. To also assess whether changes in the SAM significantly impact $nCO_2$ fluxes on an interannual timescale, we look into the detrended SAM and $nCO_2$ flux time-series (Fig. 3b and c). The correlation between the detrended SAM index and detrended $nCO_2$ flux is significant ($p \leq 0.05$) and equals 0.51 (Fig. S5a).

To assess the impact of high-resolution, and thus the lack of parametrised eddies, on the simulated $CO_2$ fluxes, the results of the $1°$ version of ACCESS-OM2 are also included (Fig. 2c, blue). If zonally-averaged, they closely track the results of ACCESS-OM2-01, displaying a similar trend and interannual variability in SO $nCO_2$ fluxes.

The simulated $aCO_2$ uptake increases by 0.56 GtC/yr over the period 1980-2021 (Fig. 2d, grey and orange) noting that the $CO_{2atm}$, which is a forcing of the model, also increases during that time (Fig. 2a). To better highlight variability in the $aCO_2$ flux, we detrend it and plot the anomaly with respect to the 1980-2021 mean (Fig. 3d). The $aCO_2$ uptake also features decadal-scale variability, but with an amplitude that is about 30% lower than for $nCO_2$. A weak but significant ($p \leq 0.05$) relationship between $aCO_2$ uptake and the detrended SAM index is simulated (R=-0.42, Fig. S5b). Increased $aCO_2$ uptake occurs when the westerlies strengthen (R=-0.36, Fig. S5e). While the $1°$ model displays a slightly larger $aCO_2$ uptake, the trends are similar between 1980 and 2018 (Fig. 2d, blue) and the detrended fluxes track each other very closely (Fig. 3d).

The combined effect of reduced $nCO_2$ uptake and increased $aCO_2$ uptake (mostly due to the increase in $CO_{2atm}$) lead to a 0.36 GtC/yr increase in $tCO_2$ uptake between 1980 and 2021 (Fig. 2e). The SO $tCO_2$ uptake in the $0.1°$ and $1°$ simulations track each other fairly closely, even though the trend is larger in the $1°$ (-0.015 GtC/yr$^2$) than in the $0.1°$ (-0.013 GtC/yr$^2$) (Fig. 2e, blue compared to grey). The simulated $tCO_2$ fluxes can be compared to the observational estimates derived from the SOM-FFN based on the SOCAT and SOCCOM biogeochemistry floats from 1982 to 2017 (red) (Landschützer et al., 2019; Bushinsky et al., 2019), and based on SOCAT only from 1982 to 2021 (magenta) (Landschützer et al., 2020) (Fig. 2g). The simulated $tCO_2$ flux and observational estimates of $tCO_2$ flux are well correlated (R=0.55 for SOCAT+SOCCOM compared to ACCESS-OM2-01 and R=0.79 for SOCAT only compared to ACCESS-OM2-01).

In the $0.1°$ experiment, the simulated $tCO_2$ uptake increases by only 0.003 GtC/yr$^2$ between 1980 and 1998 (Fig. 2e), in agreement with both observational estimates (Fig. 2f). While the simulated $tCO_2$ uptake decreases between 1998 and 2001 as in the observations, the magnitude of this simulated change is smaller than in the observational estimates. Similarly, while both simulation and observational estimates display an increase in $tCO_2$ uptake in the early 2000s, the reinvigoration only lasts until 2003 in the simulation, while it lasts until 2010 in both observational datasets. Finally, similar to the SOCAT only product, the simulation suggests a small re-invigoration of the $tCO_2$ uptake since 2012, while the SOCAT+SOCCOM product suggests a decrease in $tCO_2$ uptake. While the simulated $tCO_2$ changes are within the uncertainty range of the observational estimates ($\pm 0.15$ GtC/yr) (Bushinsky et al., 2019) for most of the simulated period, the simulated variations are lower and outside of the uncertainty range between 1998 and 2005.

Since changes in $tCO_2$ flux are also impacted by the $CO_{2atm}$ increase, we detrend the SO $tCO_2$ flux and plot the anomaly with respect to the detrended 1980-2021 mean to properly assess the decadal-scale variability. The detrended $tCO_2$ fluxes (Fig.

3e) present variations similar to the detrended $nCO_2$ fluxes (Fig. 3c), with reduced total uptake during positive phases of the SAM (Fig. 3b). The $nCO_2$ flux variability dominates the changes in $tCO_2$ uptake with a strengthening of the winds and a poleward shift both reducing the $tCO_2$ uptake (Figs. 3c,e and S5f,i). The detrended fluxes in the 0.1° and 1° simulations track each other very closely (Fig. 3e).

The correlation between detrended simulated and observationally estimated $tCO_2$ fluxes are 0.35 for SOCAT+SOCCOM (Bushinsky et al., 2019) and 0.37 for SOCAT only Landschützer et al. (2020). The two main disagreements are in the mid 1990s and the late 2000s/early 2010s, when the model simulates relatively low $tCO_2$ uptake (Fig. 3e) while the observational estimates suggest high $tCO_2$ uptake (Fig. 3f). During these two periods the detrended $nCO_2$ fluxes are small, whereas the detrended $aCO_2$ fluxes are positive. These periods of low $tCO_2$ uptake in the model are thus due to reduced $aCO_2$ uptake, 285    probably resulting from the atmospheric $CO_2$ forcing.

### 3.3    Processes leading to changes in natural $CO_2$ fluxes

#### 3.3.1    Multi-decadal trend

To better understand the processes driving the multi-decadal increase in SO $nCO_2$ outgassing, we look into the surface water natural $pCO_2$ trend and the contributions to this trend of changes in nDIC, ALK, surface temperature (SST) and salinity (SSS, 290    Fig. 4) in the ACCESS-OM2-01.

    The largest positive natural $pCO_2$ trend is simulated south of the PF and particularly south of 60°S, in the area of high upward Ekman pumping (Fig. 1e). The natural $pCO_2$ trends are particularly large in the Atlantic and Indian sectors. This is due to an increase in surface nDIC (Fig. 4b), partly compensated by an increase in ALK (Fig. 4c). We also note an area displaying a negative natural $pCO_2$ trend, off the Ross Sea and extending eastward. This is due to a decrease in nDIC, partly compensated 295    by a decrease in ALK. On the other hand, the changes in SST and SSS (Fig. 4d,e) are not contributing significantly to the multi-decadal changes in surface natural $pCO_2$.

    Taking also into account that changes in detritus flux are concentrated north of 50°S (Fig. S3k), these results suggest that changes in oceanic circulation, and particularly the upwelling strength (Fig. S3j) and subsequent northward Ekman transport (Fig. S3i), are responsible for the positive $nCO_2$ outgassing trend (Figs. 2c and S3l). The non-thermal natural $pCO_2$ changes 300    are indeed significantly correlated with changes in the strength of the westerlies in each sector of the Southern Ocean (Fig. S6a,d). The inter-basin differences in SH westerlies latitudinal trends probably contributed to the relatively larger increase in surface natural $pCO_2$ in the Atlantic and Indian sectors compared to the Pacific sector (Fig. S6e). Indeed, in the JRA55-do dataset there is a ∼1.5° and ∼1° poleward shift in the Atlantic and Indian sectors, respectively starting in the late 1990s, while there are no significant latitudinal changes in the Pacific sector (Fig. S6e). This is broadly consistent with observations, which 305    suggest a ∼1° poleward shift in the Atlantic and Indian sector and a ∼1° equatorward shift in the Pacific sector since the 1980s (Waugh et al., 2020).

    The processes leading to the positive natural $pCO_2$ trend can be further assessed by analysing the changes in SO nDIC between 1980 and 2021 (Fig. 5). An increase in nDIC is simulated within the upwelling branches of the North Atlantic Deep

Water, Indian Deep Water and Pacific Deep Water (Fig. 5a-c), while the nDIC concentration is reduced below 3000m depth, particularly in the Atlantic and Indian sectors as well as within the SAMW. The nDIC increases within the upwelling branches are mostly due to an increase in remineralized DIC (Fig. 5d-f). It is unlikely that the detritus flux increase centered at 42°S (Fig. S3k) is responsible for these positive remineralized DIC anomalies, instead they most likely indicate a higher proportion of older/deeper waters, consistent with enhanced SO upwelling.

The negative nDIC anomalies at depth are concentrated in the Antarctic Bottom Water (AABW) formation regions (Weddell Sea, Ross Sea and Adelie coast), and in the subsequent transport regions westward around Antarctica (Morrison et al., 2020; Solodoch et al., 2022). The negative nDIC anomalies at depth and within the SAMW are due to reduced remineralized DIC content, most likely implying higher transport rates of SAMW and AABW.

A reduced vertical nDIC gradient is simulated in all basins south of the SAF, due to higher nDIC in the top $\sim$1000m depth and reduced nDIC at depth. This pattern is consistent with enhanced SO upwelling resulting from a strengthening and poleward shift of the SH westerlies since 1980.

We next look at the evolution of nDIC in the subsurface of the SO between 1980 and 2021, and assess its link to the surface variability (Fig. 6). The nDIC concentration at 1000m depth gradually increases at all latitudes during that time period, but with a steepest increase between 45°S and 60°S (Fig. 6b), which corresponds to the upwelling branch of the Indian Deep Water, Pacific Deep Water and North Atlantic Deep Water. At 400m depth, nDIC also increases south of the SAF due to the enhanced upwelling of nDIC-rich deep waters (Fig. 6a). On the other hand, nDIC at 400m depth decreases north of the SAF within SAMW. This contrasting behavior north and south of the SAF could be linked to the poleward shift of the SH westerlies. At both 400m and 1000m depth, decadal-scale DIC variations are visible (Fig. 6). The fast response of subsurface DIC to the surface forcing is consistent with nDIC changes being due to Ekman pumping and associated isopycnal displacement, which respond quickly to surface forcing (Waugh and Haine, 2020).

### 3.3.2 Impact of the SAM on SO $CO_2$ fluxes

Overall, an increase in natural $CO_2$ outgassing is simulated in SO (Fig. 2c) as a response to an increase in surface nDIC. Superimposed on this trend are increases in $nCO_2$ outgassing during positive phases of the SAM (Figs. 2b,c and 3b,c). To better highlight the quantitative impact of positive phases of the SAM, we perform a composite of the years in which the annual mean SAM index as calculated from the JRA55-do dataset was greater than 0.33 (i.e. 1998, 1999, 2010, 2015 and 2021) and compare this to a composite of negative SAM years (SAM index$\leq$-0.33: 1980, 1991, 1992, 2002).

During these strong positive phases of the SAM, a significant increase ($\geq$1 mol/m$^2$/yr) in $nCO_2$ outgassing is simulated south of the PF (Fig. 7a). Some enhanced $nCO_2$ outgassing is also simulated between the SAF and PF in the Indian and southwest Pacific sector as well as north of the SAF in the Pacific sector. This $nCO_2$ outgassing mostly results from an increase in surface nDIC concentration (Fig. 7b, 9b).

The stronger and poleward shifted westerlies during positive phases of the SAM enhance the Ekman-driven vertical DIC advection south of the PF (Figs. 7g and 8b). The associated deepening of the mixed layer also drives an increase in vertical DIC diffusion at the base of the mixed layer (Figs. 7e,f and 8c). South of the PF, the Ekman-driven vertical DIC advection

and vertical diffusion at the base of the mixed layer contribute equally to the DIC increase (2.8 GtC/yr, Fig. 8b,c). The eddy-driven vertical DIC advection (taken as the difference between the vertical DIC advection and the Ekman-driven vertical DIC advection) further contributes to the higher surface DIC south of 60°S (+0.7 GtC/yr, not shown). However, north of the PF, the Ekman-driven vertical DIC advection decreases surface DIC, while vertical diffusion at the base of the mixed layer leads to a DIC increase in the mixed layer. In this simulation, changes in biological export of carbon are two orders of magnitude smaller than the Ekman-driven and vertical diffusion contributions and therefore do not affect significantly changes in $nCO_2$ fluxes (Fig. 8d).

Due to the long-term shift towards positive phases of the SAM, the composite of positive SAM is shifted towards more recent years than the composite of negative phases. If we correct for this (i.e. assuming that the $aCO_2$ uptake follows a linear trend), then an anomalous $aCO_2$ uptake is simulated south of the PF (Fig. 7c), in regions where a stronger $nCO_2$ outgassing is simulated (Fig. 7a). The amplitude of the $aCO_2$ anomalies are however only equivalent to ~25% of the $nCO_2$ anomalies (Figs. 7a,c and 8e,f black line). As a result, reduced $tCO_2$ uptake is simulated south of the PF during positive phases of the SAM (Fig. S7b). If a similar correction is applied to compensate for the difference in mean year between the positive and negative SAM composites, then an anomalous $tCO_2$ outgassing is simulated almost everywhere in the SO (Figs. 7d and 8g, black line).

It is interesting to note that south of the PF, the regions of maximum $nCO_2$ outgassing and $aCO_2$ uptake averaged over the period 1982-2021 (Fig. 1c,d) are similar to the maximum $nCO_2$ outgassing and $aCO_2$ uptake anomalies obtained for positive phases of the SAM (Fig. 7a,c), indicating that the positive phases of the SAM simply accentuate the mean SO features (i.e. $nCO_2$ outgassing gets stronger in outgassing regions). Some of the main areas of $nCO_2$ outgassing and $aCO_2$ uptake correspond to major topographic features of the SO: namely, the eastern part of the Southeast Indian Ridge, Drake Passage and the Southwest Indian Ridge (Fig. 1c,d,e). On the contrary, the cyclonic circulation in the relatively deep basin of the eastern part of the Ross Sea and Amundsen-Bellinghausen Sea is associated with $nCO_2$ uptake. This could be due to enhanced eddy mixing over topography linked to the merging of multiple jets (Lu and Speer, 2010), and warrants further study.

## 4  Discussion and conclusions

We have used an eddy-rich global ocean, sea-ice, carbon cycle model to assess changes in SO total, natural and anthropogenic $CO_2$ fluxes over the last 50 years. The multi-decadal strengthening and poleward shift of the SH westerlies, associated with a shift towards positive phases of the SAM during that period, drives a decrease in $nCO_2$ uptake with a trend of -0.007 GtC/yr$^2$. On the other hand, the increase in $CO_{2atm}$ growth rate leads to a higher $aCO_2$ uptake with a trend of 0.014 GtC/yr$^2$. A strengthening and poleward shift of the SH westerlies enhance the $aCO_2$ uptake but the magnitude of this change is only 30% of the associated enhanced $nCO_2$ outgassing. As a result, while the $tCO_2$ uptake increases between 1980 and 2021 with a trend of 0.007 GtC/yr$^2$, it would have likely increased twice as fast without a strengthening and poleward shift of the SH westerlies. These $CO_2$ flux trends simulated with a high-resolution eddy-rich model are similar to those obtained by a similar simulation performed with ACCESS-OM2 at 1° resolution (Fig. 2), even though the $nCO_2$ trend is slightly smaller (-0.005 GtC/yr$^2$) and the $tCO_2$ trend larger (0.009 GtC/yr$^2$) in the 1° than the 0.1° experiment. These results are also consistent with those of

Lovenduski et al. (2008), who simulated an increase in $nCO_2$ outgassing between 1979 and 2004 with a trend of 0.004 GtC/yr$^2$, an increase in $aCO_2$ uptake with a trend of 0.011 GtC/yr$^2$ and thus an increase in $tCO_2$ uptake of 0.007 GtC/yr$^2$ using a coarse resolution ocean model. The multi-decadal, large-scale oceanic carbon cycle response to a strengthening and poleward shift of the SH westerlies is thus robust from eddy-rich to coarse resolution models.

In addition, the total air-sea $CO_2$ fluxes exhibit large ($\sim$0.1 GtC/yr) decadal-scale variability thus supporting previous inferences of decadal scale changes in SO $CO_2$ fluxes (Li and Ilyina, 2018; Lovenduski et al., 2008; Landschützer et al., 2015; Gruber et al., 2019). The simulated variability is not as large as that derived from observational estimates ($\sim$0.25 GtC/yr) (Landschützer et al., 2016; Bushinsky et al., 2019; Keppler and Landschützer, 2019), but is within the uncertainty band ($\pm$0.15 GtC/yr) (Gruber et al., 2019; Bushinsky et al., 2019). Such a mismatch between simulated SO $tCO_2$ variations and observations is prevalent in hindcast simulations (Gruber et al., 2019; Hauck et al., 2020) and could be due to an overestimation of the observed SO $CO_2$ flux variability (Gloege et al., 2021). The underestimation of the changes in $tCO_2$ uptake in the simulation could also be due a misrepresentation of Southern Ocean stratification. It has indeed been suggested that the overturning rate of the lower cell weakened in the 2000s (DeVries et al., 2017) due to enhanced stratification in the Southern Ocean (de Lavergne et al., 2014), linked to enhanced Antarctic basal melt rates (Adusumilli et al., 2020). Enhanced stratification in the Southern
Ocean would weaken the $aCO_2$ uptake (Bourgeois et al., 2022), but would reduce the $nCO_2$ outgassing (Menviel et al., 2015), thus potentially enhancing $tCO_2$ uptake.

To first order, the simulated decadal-scale changes in $tCO_2$ fluxes are due to changes in $nCO_2$ fluxes primarily arising from changes in the magnitude of the SH westerlies, but also due to variations in the latitudinal position of the SH winds. While we find a strong link between regional wind changes and $nCO_2$/$tCO_2$ fluxes, we find that minima in $tCO_2$ uptake arise from
a strengthening and/or poleward shift of the SH westerlies, and thus positive phases of the SAM. This is in contrast to the conclusion of Keppler and Landschützer (2019) that the SAM had a net zero effect on SO $tCO_2$ uptake. Both our study and the one of Keppler and Landschützer (2019) highlighted enhanced $tCO_2$ outgassing south of $50°$S during positive phases of the SAM as well as zonal asymmetries with enhanced $tCO_2$ uptake in the Pacific sector of the SO. While Keppler and Landschützer (2019) suggest this is linked to the zonal wave number 3 pattern, we attribute these asymmetries to the bathymetry and different
poleward trends of the westerlies in the different sectors of the SO.

A stagnation of SO $tCO_2$ uptake between 1980 and 2000 is simulated. This time period corresponds to the largest rate of increase and shift in westerly wind stress. The timing and magnitude of this stagnation in $tCO_2$ uptake in the SO is in agreement with observational estimates (Lovenduski et al., 2008; Landschützer et al., 2015; Gruber et al., 2019; Keppler and Landschützer, 2019). While the impact of the SAM on SO $CO_2$ fluxes is clear in our simulation, the early 1990s also feature
the lowest atmospheric $CO_2$ growth rate of the period studied here (McKinley et al., 2020). The simulated SO $aCO_2$ uptake in the early 1990s is thus the lowest of the period, noting that positive phases of the SAM are usually associated with slightly enhanced $aCO_2$ uptake. Our results thus also support the conclusion that the slowdown of the SO $tCO_2$ uptake in the early 1990s was due to a low atmospheric $CO_2$ growth rate (McKinley et al., 2020) and not a positive phase of the SAM (LeQuéré et al., 2007; Lovenduski et al., 2008; Matear and Lenton, 2008). In agreement with observations, a re-invigoration of $tCO_2$ uptake
is simulated in the early 2000s (Keppler and Landschützer, 2019), mostly due to a pause in the positive SAM trend. Since the

mid 2000s, the $tCO_2$ uptake has increased slowly, but we find that the reversal in $tCO_2$ uptake that had been highlighted in the mid 2010s (Keppler and Landschützer, 2019) was short-lived and due to the strong positive 2015 SAM.

The enhanced $nCO_2$ outgassing during positive phases of the SAM is due to higher surface nDIC concentration south of the PF, partly compensated by lower SST. This increase in surface nDIC results from enhanced vertical nDIC advection, mostly Ekman driven, as well as enhanced vertical nDIC diffusion at the base of the mixed layer. This significant role of vertical diffusion is in agreement with a previous study performed with an eddy-permitting model (Dufour et al., 2013), even if contrarily to that study we find an equal weight of vertical advection and Ekman pumping south of the PF. The dominance of Ekman driven vertical nDIC advection also explains the similar results obtained in the high and coarse resolution versions of the ACCESS-OM2. As in previous studies, we thus find that changes in oceanic circulation are the primary driver of changes in SO $CO_2$ fluxes on decadal-time scales (Dufour et al., 2013; Resplandy et al., 2015; Nevison et al., 2020).

Previous studies have suggested that wind-driven changes in oceanic circulation in the Southern Ocean are partially compensated by the eddy-driven transport (Morrison and Hogg, 2013). Similarly, Dufour et al. (2013) suggested that 1/3 of the Ekman-driven DIC transport arising from positive phases of the SAM was compensated by eddy transport. Here, despite a 20% increase in wind stress, only a small ACC increase (134 to 138 Sv) is simulated thus supporting the eddy saturation theory. Yet, we find that changes in the position and strength of the SH westerlies lead to an outgassing of $nCO_2$ on a yearly as well as multi-decadal timescale, with an amplitude similar to that found in a similar model (ACCESS-OM2) with a 1° resolution. This is an important result as it was suggested that mesoscale eddies would compensate for the wind-driven circulation in the Southern Ocean, thus mitigating the carbon cycle response to changes in the strength and position of the westerlies. Here we show that even in an eddy-rich model, a strengthening and/or a poleward shift of the westerlies leads to enhanced $CO_2$ outgassing. This further suggests that ocean models with a $\sim$1° resolution correctly capture the large-scale carbon cycle response to changes in the SH westerlies. It should however be noted that in the 1° ocean model used here, the GM coefficient varies in space and time (Kiss et al., 2020). While both the 0.1° and 1° resolution simulations display broadly similar mean $CO_2$ fluxes (Figs. 1 and S4) and $CO_2$ fluxes response to positive phase of the SAM (Figs. 7 and S8), higher $nCO_2$ outgassing is simulated south of the PF in the 0.1° than 1° resolution. This could be due to larger upwelling downstream of topographic features in the eddy-rich version of the model.

If SH westerly winds continue to strengthen, as projected under RCP8.5/SSP5-85 scenarios (Grose et al., 2020; Goyal et al., 2021), our experiments suggest that the increase in $aCO_2$ uptake would be partly compensated by $nCO_2$ outgassing, thus leading to only a small increase in $tCO_2$ uptake. Future changes in SO carbon uptake will thus likely result from a fine balance between natural carbon release and anthropogenic carbon uptake, which will itself depend on changes in SH westerlies, SO stratification and temperature as well as the rate of anthropogenic carbon emissions.

*Data availability.* The data linked to this study has been deposited on UNSWorks http://hdl.handle.net/1959.4/101483 under the doi https://doi.org/10.2619

*Acknowledgements.* This project was supported by the Australian Research Council (ARC), including grants FT180100606, FT190100413, and SR200100008. AEK was supported by ARC grants LP160100073 and LP200100406, and the Australian Government's Australian Antarctic Science Program grant 4541.

The authors thank the Consortium for Ocean-Sea Ice Modelling in Australia (COSIMA; http://www.cosima.org.au) for making the ACCESS-OM2 suite of models available at https://github.com/COSIMA/access-om2. Model runs were undertaken with the assistance of resources from the National Computational Infrastructure (NCI), which is supported by the Australian Government. The authors thank Kial Stewart for sharing the code to calculate the SAM index from the JRA-55do dataset.

**Author contribution** LM designed the study with PS and DW. AEK, MAC and HH developed some parts of the model

and ran the experimet in collaboration with the COSIMA consortium. LM analysed the data and wrote the manuscript with contributions from all authors.

**Competing interests** The authors have no competing interests.

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

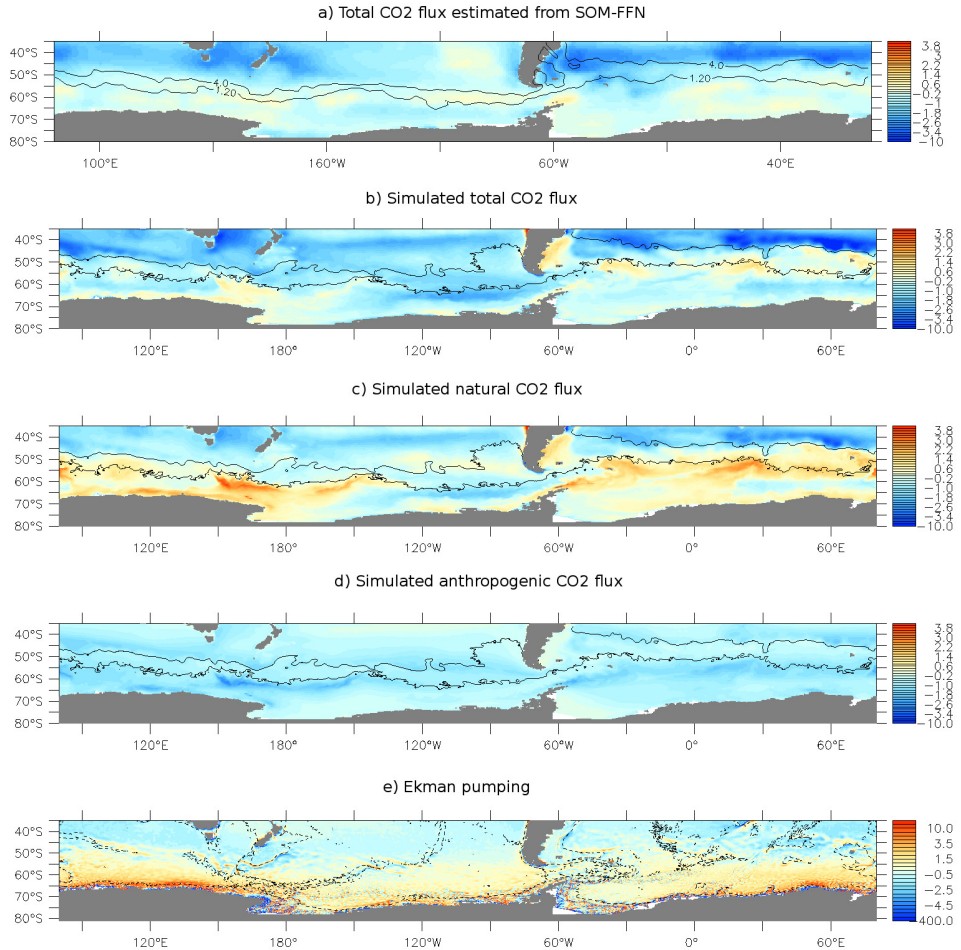

**Figure 1.** a) Total ocean to atmosphere $CO_2$ flux (molC/m$^2$/yr) as estimated from the SOM-FFN for the period 1982-2021 (Landschützer et al., 2016, 2020). The black contours indicate (from south to north) the northern edges of the polar front (PF) and the sub-Antarctic front (SAF) using the definition of Sokolov and Rintoul (2009) and the temperature data from the World Ocean Atlas (Locarnini et al., 2013). Simulated b) Total, c) natural and d) anthropogenic ocean to atmosphere $CO_2$ flux (molC/m$^2$/yr) averaged over the period 1982-2021 in the 0.1° ACCESS-OM2-01. The black contours indicate the northern edges of the PF and SAF using the definition of Sokolov and Rintoul (2009). e) Ekman pumping (x10$^{-6}$ m/s) averaged over the period 1982-2021 in the numerical experiment. The black lines overlaid represent the 2500m depth bathymetry contour.

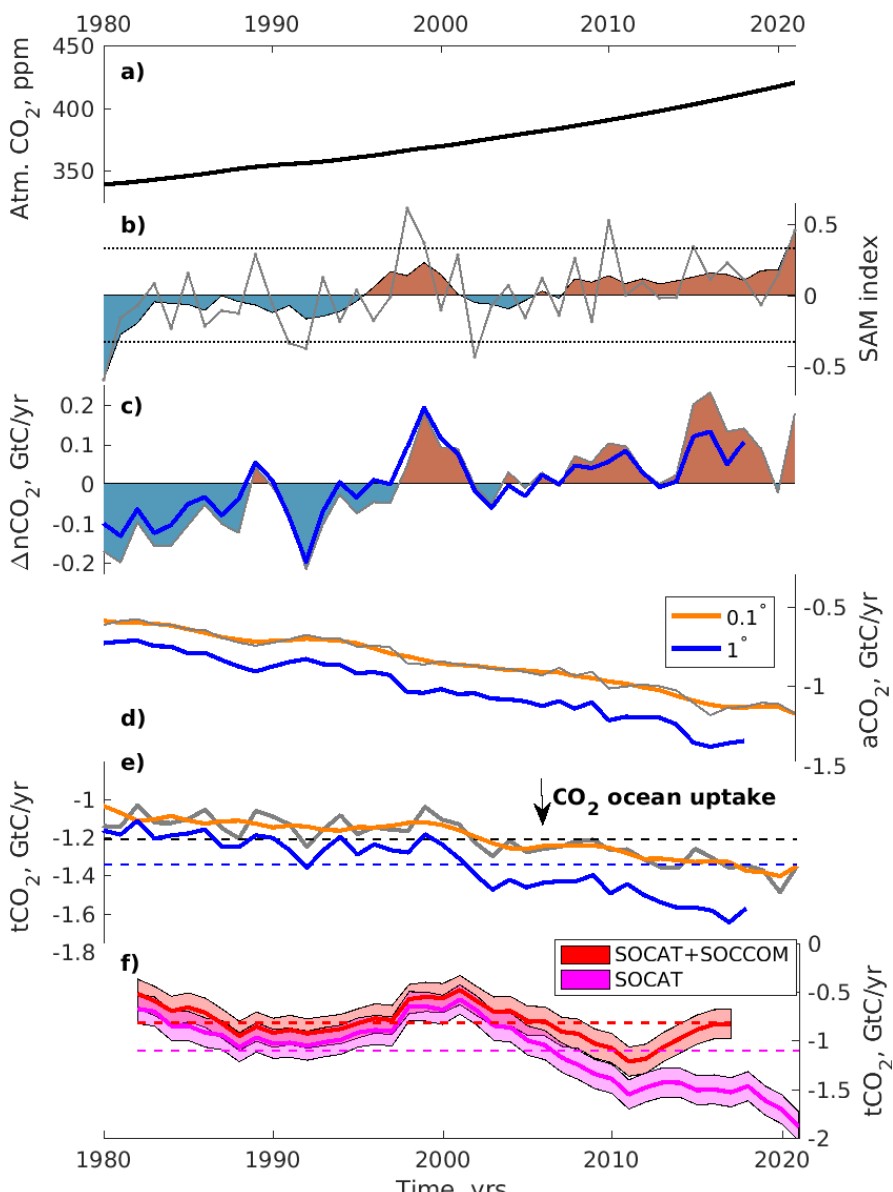

**Figure 2.** Time series of a) annual mean atmospheric $CO_2$ concentration used as forcing, b) SAM index calculated from the JRA55-do dataset (Stewart et al., 2020). The horizontal dotted lines represent the thresholds used to define positive and negative SAM in the composites. Simulated integrated ocean to atmosphere $CO_2$ fluxes in the ACCESS-OM2-01 (0.1°, annual mean in grey and 5-yr running mean in orange) and ACCESS-OM2 (1°, annual mean in blue) simulations: c) $nCO_2$, d) $aCO_2$, and e) $tCO_2$. f) SO $tCO_2$ flux as derived from the SOM-FFN (red) including both the SOCAT and SOCCOM data (Bushinsky et al., 2019), and (magenta) only including the SOCAT data (Landschützer et al., 2020). The shading represents an uncertainty of 0.15 GtC/yr. All the $CO_2$ fluxes are integrated over the SO (35°S-80°S) and are in GtC/yr. Dashed horizontal lines represent the mean over 1980-2021 or over the available period.

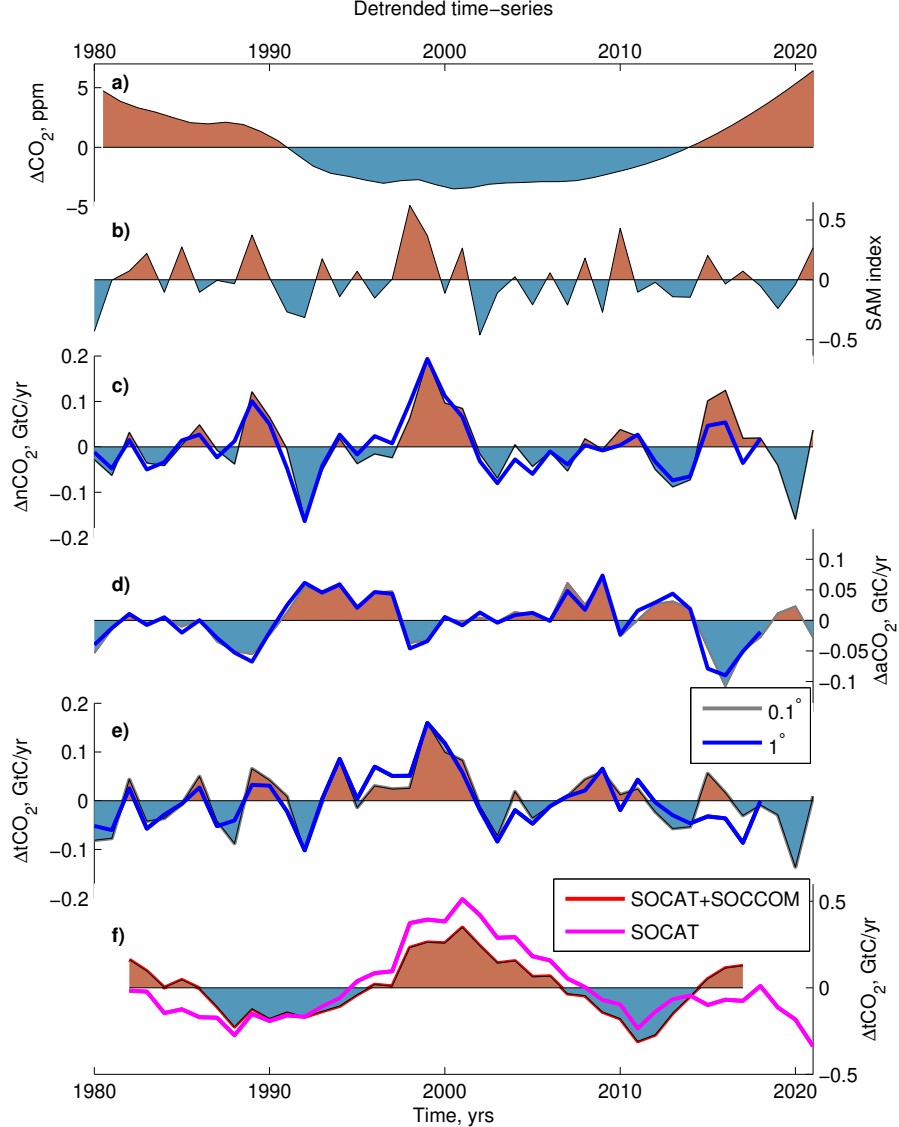

**Figure 3.** Detrended annual mean time series of a) atmospheric $CO_2$ (ppm) used as forcing, b) SAM index calculated from the JRA55-do dataset (Stewart et al., 2020). Detrended annual-mean simulated integrated ocean to atmosphere annual mean detrended $CO_2$ fluxes in the ACCESS-OM2-01 (0.1°, grey, with light blue and red shadings indicating positive and negative anomalies with respect to the mean) and the ACCESS-OM2 (1°, blue) simulations: c) $nCO_2$, d) $aCO_2$, e) $tCO_2$. f) Detrended SO $tCO_2$ flux as derived from the SOM-FFN including both the SOCAT and SOCCOM data (red) (Bushinsky et al., 2019), and including the SOCAT data only (magenta) (Landschützer et al., 2020). All the $CO_2$ fluxes are integrated over the SO (35°S-80°S) and are in GtC/yr.

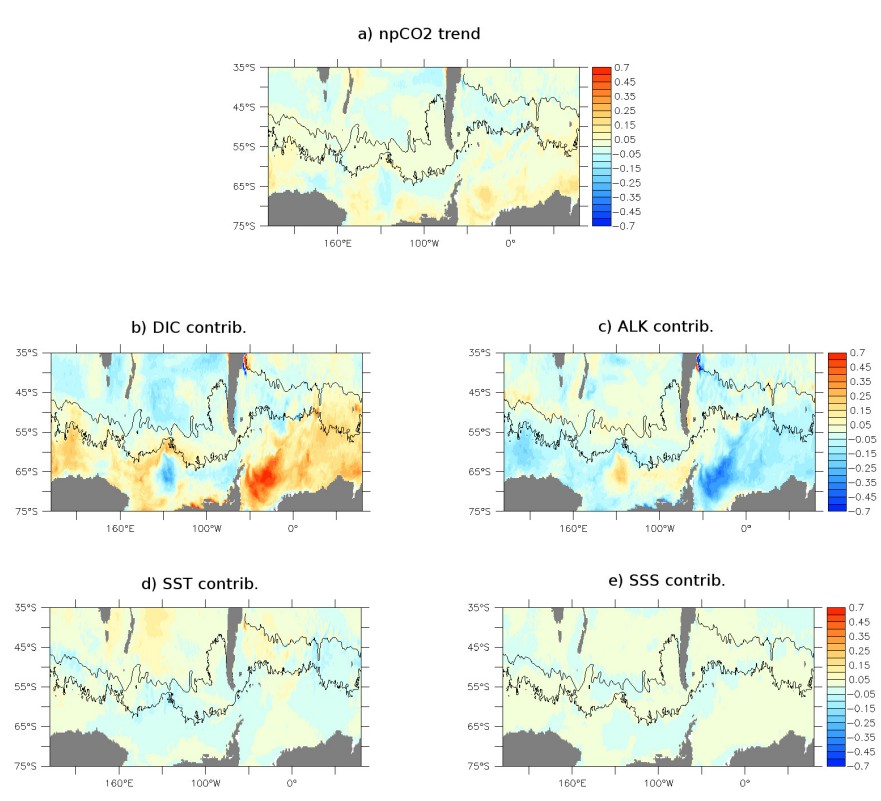

**Figure 4.** a) Natural pCO$_2$ trend (ppm/decade) between 1981 and 2021 and b) nDIC, c) ALK, d) SST and e) SSS contributions to the natural pCO$_2$ trend (ppm/decade) for the ACCESS-OM2-01 simulation.

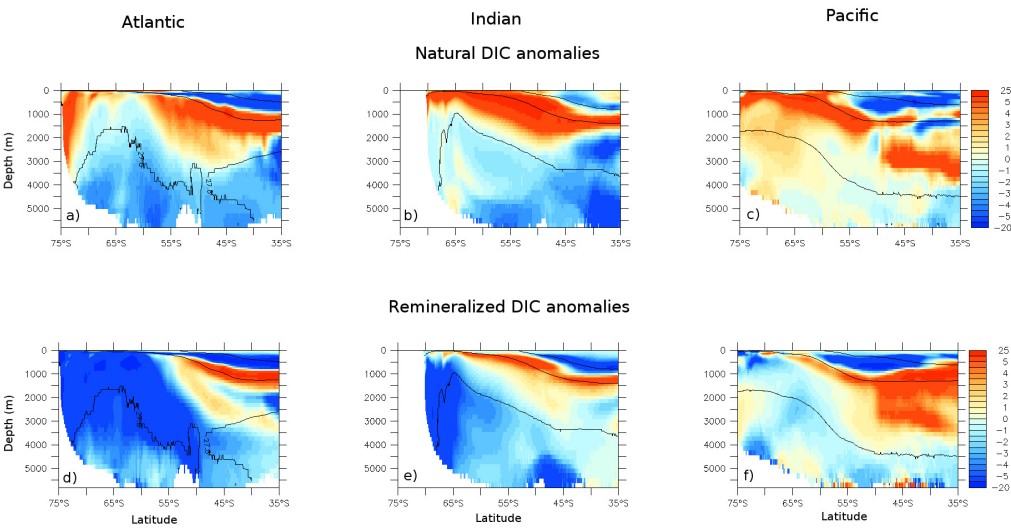

**Figure 5.** Zonally averaged (a-c) natural DIC, and (d-f) remineralized DIC (mmol/m$^3$) averaged over (left) the Atlantic, (middle) the Indian and (right) the Pacific for years 2017-2021 compared to 1980-1982 for the ACCESS-OM2-01 simulation. The density of the AABW ($\geq$1028.31 kg/m$^3$), the AAIW (1027.5$\geq$AAIW$\geq$1026.95 kg/m$^3$) and the SAMW ($\leq$1026.95 kg/m$^3$) are overlaid.

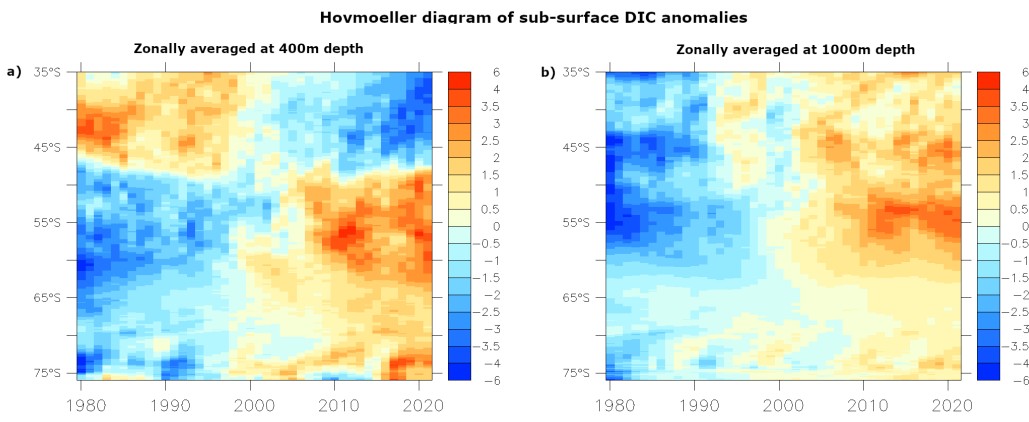

**Figure 6.** Hovmoeller diagram over the period 1980-2021 of zonally averaged nDIC anomalies (mmol/m$^3$) as a function of time and latitude at a) 400 m and b) 1000 m depth compared to the time average nDIC in the ACCESS-OM2-01 simulation.

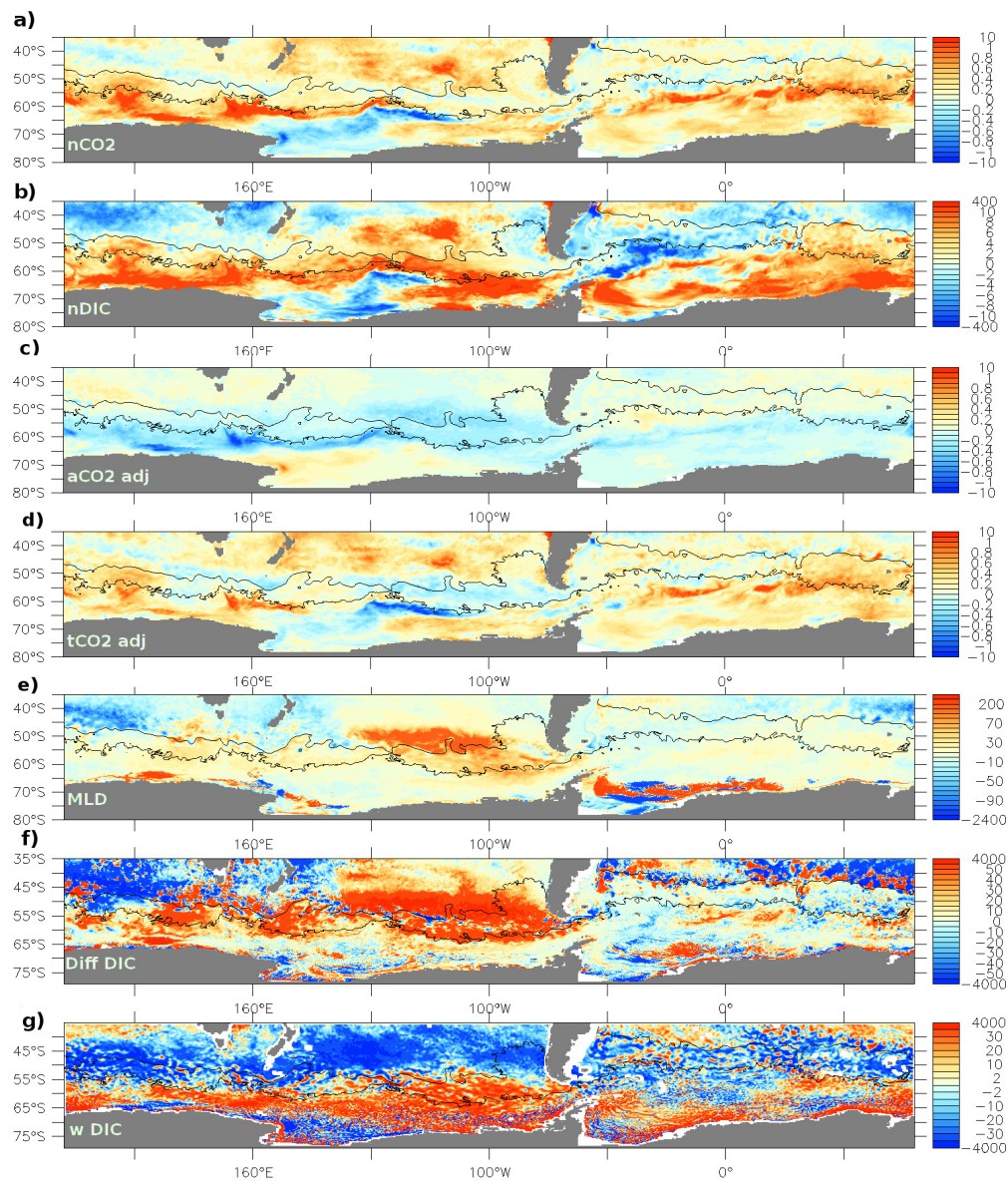

**Figure 7.** a) $nCO_2$ flux (mol/m$^2$/yr), b) surface nDIC (mmol/m$^3$), c) adjusted $aCO_2$ flux (molC/m$^2$/yr) and d) adjusted $tCO_2$ flux (molC/m$^2$/yr) anomalies for a composite of positive phases of the SAM ($\geq 0.33$, i.e. 1998, 1999, 2010, 2015 and 2021) compared to a composite of negative SAM years ($\leq 0.33$, i.e. 1980, 1991, 1992, 2002) for the ACCESS-OM2-01 simulation. Linear trends in $aCO_2$ fluxes have been removed from the $aCO_2$ and $tCO_2$ anomalies to take into account the difference in mean years between the composite of positive and negative SAM years. Annual average anomalies of e) maximum monthly mixed layer depth (m), f) vertical diffusivity multiplied by the DIC gradient at the base of mixed layer (molC/m$^2$/yr) and g) vertical Ekman DIC advection with a 21 point spatial smoothing (molC/m$^2$/yr), for positive phases of the SAM compared to negative SAM years.

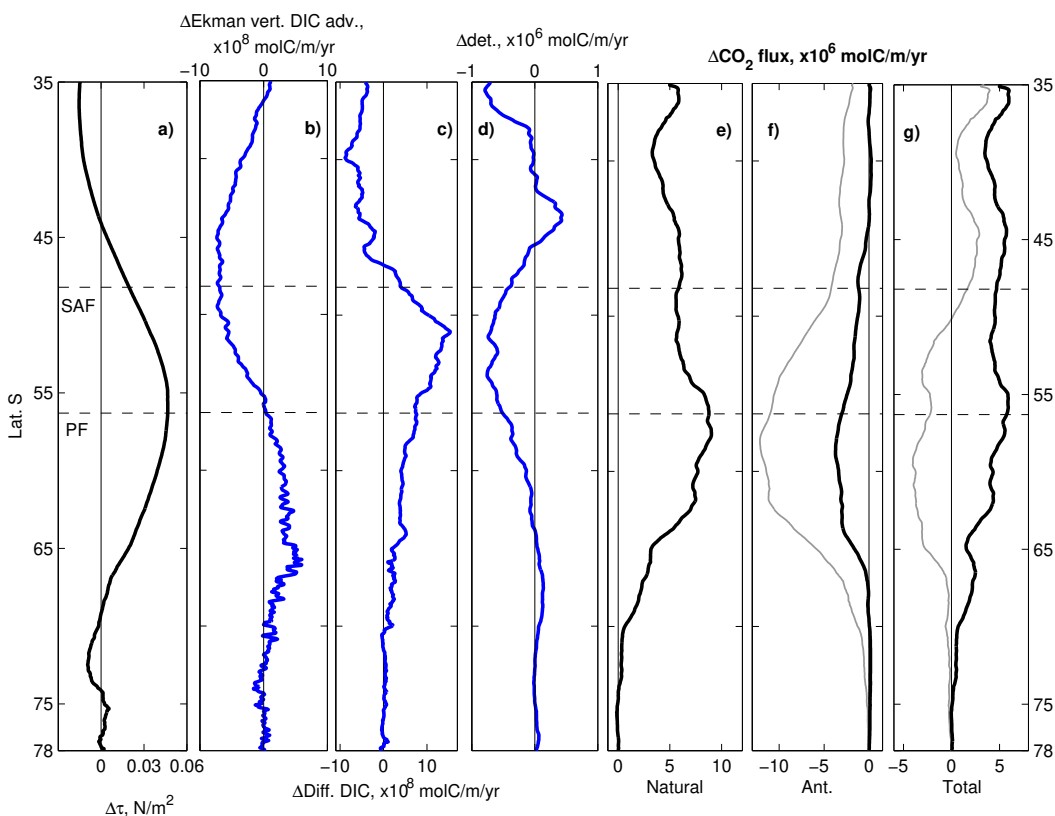

**Figure 8.** a) Zonally averaged wind stress anomalies (N/m$^2$); Anomalies in zonally integrated b) vertical Ekman DIC advection (x10$^8$ molC/m/yr), c) vertical diffusivity multiplied by the DIC gradient at the base of mixed layer (x10$^8$ molC/m/yr), d) detritus flux at 100m depth (x10$^6$ molC/m/yr), e) nCO$_2$, f) aCO$_2$ and g) tCO$_2$ fluxes (x10$^6$ molC/m/yr) for the positive SAM composite compared to the negative SAM composite in the ACCESS-OM2-01 simulation. In f and g) the grey lines represent the simulated aCO$_2$ and tCO$_2$ fields while the black lines include a correction for the fact that the positive SAM composite represents more recent years than the negative SAM composite. The linear trends in aCO$_2$ and tCO$_2$ fluxes between 1980 and 2021 are calculated. The equivalent mean aCO$_2$ and tCO$_2$ flux differences between the mean positive and negative SAM composites are then subtracted.

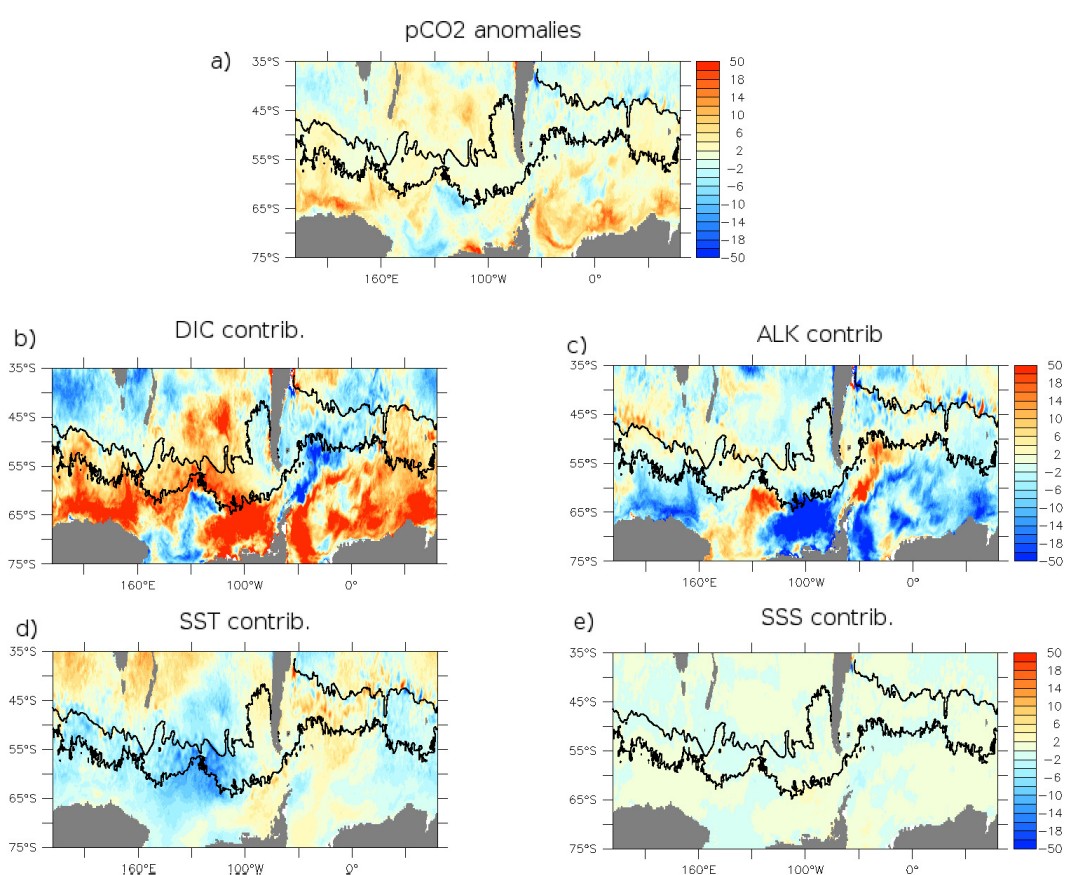

**Figure 9.** Surface ocean a) natural pCO₂ anomalies (ppm) for a composite of positive phases of the SAM compared to a composite of negative SAM years (see Fig. 7) and the pCO₂ contributions (ppm) from b) nDIC, c) ALK, d) SST and e) SSS for the ACCESS-OM2-01 simulation.