# Peer review of "Enhanced Southern Ocean CO2 outgassing as a result of stronger and poleward shifted southern hemispheric westerlies"

_EGUsphere, 2023_

## Author Comment (AC1)

Reviewer 1:

This study focuses on understanding the interannual and decadal variability in Southern Ocean CO2 fluxes and their links to the Southern Annular Mode (SAM). This research is particularly significant because few numerical modeling studies investigate anthropogenic and natural carbon fluxes separately. As a result, this study is unique, aligns well with the scope of the journal, and makes a meaningful contribution to the existing literature.

Throughout the manuscript, I encountered difficulties following the connection between the figures and the text (see specific comments below). Additionally, certain statements in the manuscript are difficult to comprehend within the context provided (see specific comments below). Revising some of these points to create a more concise and coherent format would greatly benefit the manuscript.

Moreover, the authors emphasize that their study differs significantly from Lovenduski et al. (2008) due to the use of a high-resolution model. As a reader, when going through the introduction, I anticipated a more extensive discussion and conclusion section addressing the impact of using eddy-resolving models. Additionally, I expected to see some recommendations as a conclusion. However, this was not the case, as it was only briefly mentioned in one or two sentences in the discussion section. I suggest expanding the discussion on the effects of using high-resolution models in such studies, even in a broader context.

We thank the Reviewer for their helpful comments, which helped improve the manuscript. We are now connecting more clearly the text and figures, and have expanded the discussion and conclusion to highlight the use of an eddy-rich model. We have significantly modified the text and figures to include i) an improved and clearer comparison with observations, ii) an improved presentation of both full and detrended results, and iii) the inclusion of a similar simulation performed with the 1 degree version of the model.

We are answering all the Reviewer's comments below in blue, with suggested changes to the text in green.

**Specific comments and minor changes**

**Methods:**

Line 94: "has many improvements"

Could you provide more specific details regarding the improvements in the model? Additionally, please explain why the inclusion of "ocean biogeochemistry with two-way coupling to nutrient and algae carried in the sea ice model" is important for this study?

We are now adding more details on the main improvements that were made to the model from the version described in Kiss et al., 2020, and provide reference to the manuscript, which describes in more details the improvements (Solodoch et al., 2022).

That section now reads:

"ACCESS-OM2 is described in detail in Kiss et al. (2020), but the version presented here has many improvements as described in Solodoch et al., (2022). The main improvements relevant to this study are that the wind stress calculation now uses relative velocity over both ocean and sea ice (not just ocean), and the albedo of the ocean is now latitude-dependent following Large & Yager (2009)."

The sentence related to the nutrient and algae, which is an improvement of the model, was moved to the description of WOMBAT.

"This version also includes a two-way coupling of the ocean biogeochemistry with nutrient and algae carried in the sea ice model (Hayashida et al., 2021)."

Line 106: "The air-sea $CO_2$ exchange is a function of ….."

When reading this sentence, it seems to suggest that there is no effect of DIC concentration. However, this is not true. The authors also mention in the results (line 228, line 258-259) that outgassing primarily results from an increase in surface nDIC concentration.

This is amended as:

"The air-sea $CO_2$ exchange is a function of the difference in partial pressure of $CO_2$ at the air-sea interface, the wind speed (Wanninkhof et al., 1992) and sea ice concentration."

Line 132: "To better understand ….."

With the limited description provided, it is challenging to comprehend the method employed by the authors. They reference a substantial book that may not be accessible to everyone. As a result, a more detailed explanation of the equations (1 and 2) is essential in this manuscript.

We have now added some text to explain the derivation of the equations as follow:

"Oceanic natural $pCO_2$ is a function of nDIC, alkalinity (ALK) as well as ocean temperature and salinity.

Changes in $pCO_2$ can thus be described as:

$$\Delta pCO_2 = \frac{\partial(pCO2)}{\partial(DIC)}\Delta DIC + \frac{\partial(pCO2)}{\partial(ALK)}\Delta ALK + \frac{\partial(pCO2)}{\partial(Sal)}\Delta Sal + \frac{\partial(pCO2)}{\partial(T)}\Delta T$$

To better understand the processes leading to pCO2 changes, we can estimate the $pCO_2$ change from each of the above variables separately. Broecker et al., (1979) derived that if ALK, salinity and temperature are constant then:

$$\frac{\partial ln(pCO2)}{\partial\, ln\,(DIC)} = \gamma DIC$$

With $\gamma DIC$ being the Revelle factor of DIC.

Equation 1 can be re-written as:

$$\frac{DIC}{pCO2} \cdot \frac{\partial(pCO2)}{\partial(DIC)} = \gamma DIC$$

One can then derived the $pCO_2$ change due to a change in DIC ($\Delta pCO_2 dic$) as:

$\Delta pCO2_{DIC} = \gamma DIC \ pCO_2 \ \Delta DIC / DIC$,

Here we use a mean high latitude estimate of $\gamma DIC$ of 13.3 (Gruber & Sarmiento, 2006) to estimate $\Delta pCO2 dic$.

$pCO_2$ sensitivities to ALK and salinity can be derived with similar equations:

$\Delta pCO2_{alk} = \gamma ALK \ pCO2 \ \Delta ALK / ALK$,

$\Delta pCO2_{Sal} = \gamma Sal \ pCO2 \ \Delta Sal / Sal$,

With $\gamma ALK$ of -12.6, and $\gamma Sal$ of 1 (Gruber & Sarmiento, 2006).

Finally, Takahashi et al., (1993) suggest that the $pCO_2$ sensitivity to temperature (T) follows the relationship:

$\frac{\partial ln pCO2}{\partial T} \sim 0.0423 \ °C^{-1}$

This implies (Takashi et al., 2002 & 2009) that the change in $pCO_2$ due to temperature is:

$\Delta pCO2_T = (e^{(0.0423*\Delta T)} - 1) \ pCO2$

**Results:**

Line 154-155:

"tco2 fluxes can be compared to observational ….."

A more detailed description of the observations and the model, along with quantification, is needed. I expect the authors to present a comparison, explaining their approach and how the models and observations compare. Furthermore, the manuscript contains a statement about being in agreement with observations. How and where can a reader verify this? If applicable, please clarify the connection to the relevant figures.

To avoid confusion, we want to make clear that, as per its title, this section compares mean simulated and observationally-derived $CO_2$ fluxes, i.e. the longest time-average covered by both simulation and data. Time-varying fluxes are then described in section 3.2.

The self-organizing map-feed-forward neural network (SOM-FFN) provides observational estimates of total $CO_2$ flux for years 1982-2021 (Landschutzer et al., 2019). To appropriately assess the mean model performances, simulated tCO2 fluxes (Fig. 1b) are compared to the time-average of the SOM-FFN observational estimates (Fig. 1a). To make this clearer, we are

now adding more information about the observational estimates as well as a more detailed comparison between the mean simulated fluxes and estimates.

The paragraph starting L.153 now reads:

"We first assess the performances of the model by comparing the time-mean simulated SO $tCO_2$ fluxes to observational estimates (Fig. 1a,b). The Surface Ocean $CO_2$ ATlas version 6 (SOCATv6) [Bakker et al., 2016] provides surface ocean $CO_2$ measurements. However, due to the spatial and temporal heterogeneity of these measurements, it does not provide an appropriate dataset for a comparison with simulated fields. To fill this gap, Landschutzer et al., (2016) developed a method to provide a global gridded monthly observational estimate. The ocean is first clustered into biogeochemical provinces using Self-Organizing Map (SOM). Then, within each biogeochemical province, $pCO_2$ estimates are generated based on a non-linear relationship between the SOCATv6 observations and the $CO_2$ driver variables through a feed-forward neural network (FFN) approach.

If averaged over the available period of 1982-2021, the observationally-derived SOM-FFN dataset (Landschutzer et al. 2020, Fig. 1a) displays a strong $tCO_2$ uptake north of 50S (-1.59 mol/m$^2$/yr, zonal averaged between 50S and 35S) and a weak $tCO_2$ uptake (-0.38 mol/m$^2$/yr) south of 50S, even though there are some areas with outgassing (~0.2 mol/m$^2$/yr) south of 50S.

These features are relatively well reproduced by the simulated $tCO_2$ fluxes (Fig. 1b), which display a similar strong uptake (-1.59 mol/m$^2$/yr) north of 50S, that is north of the SAF (Sokolov et al. 2009). As in the observations, some $tCO_2$ outgassing is simulated south of the SAF, but particularly south of the PF. While both observational estimates and simulation suggest a $tCO_2$ outgassing south of the PF at 0-60E, 150E-180E and downstream of the Drake passage, the simulated tCO2 outgassing is particularly confined to some hotspots, namely over the eastern part of the Southeast Indian Ridge, east of the Drake Passage and over the Southwest Indian Ridge (Fig. 1b). Overall, a similarly weak $tCO_2$ uptake (-0.59 mol/m$^2$/yr) is estimated south of 50S."

**Line 164:**

"While simulated nco2 ….."

What is the reason for the sudden comparison of these two specific time periods (1980-1984 and 2017-2021)?

In the new Figures 5 and S3, we compare the end of the simulation with the beginning of the simulation. This highlights the impacts of the long-term positive SAM trend on the SO carbon cycle. The text has been modified to clarify this point.

**Line 167:**

"Through Ekman transport ….."

There seems to be no link between this statement and the figures included in the paper, so I suggest establishing a connection with the appropriate figure. Moreover, the manuscript first mentions the impact of phytoplankton on DIC here. I recommend elaborating on this effect in the methods section. If the effect is substantial, a more detailed description of the biogeochemical model (WOMBAT here) and a summary figure would be beneficial.

We are now adding a meridional Ekman transport into the new figure S3 as well as the zonally integrated detritus flux at 100m depth. The text now reads:

"Through Ekman transport, surface waters in the SO move equatorward (Fig. S3b), and nutrients and DIC are consumed by phytoplankton, leading to a maximum detritus flux at ~42S (Fig. S3d) and $nCO_2$ ocean uptake north of the SAF (Fig. S3e), where Antarctic Intermediate Waters (AAIW) and Subantarctic Mode Waters (SAMW) are formed."

**Line 171:**

"The nco2 ….. "

Could you please provide a reference to a particular figure where this information can be easily followed?

Figure 2c is mentioned L. 170. To make it clearer, we are now also referring to Fig. 2c in the sentence starting L. 171 and finishing L. 172.

**Line 171:**

"While this is compared SAM index ….. "

The comparison made by the authors is not evident within the paper, requiring readers to consult the Marshall 2003 paper. For improved clarity, could you add Marshall's data to the relevant figure and mention that figure in this statement?

We have removed the reference to Marshall et al. 2003 in that sentence, but have added a sentence in the Methods about the agreement between the SAM index derived from the JRA55-do dataset and the one of Marshall et al., 2003.

**Line 178:**

"The nco2 ….. "

The authors suddenly give a spatial pattern. What is the reason for this? It does not appear to be relevant to this subsection.

We understand this might not be the best location for this sentence, which was thus removed here.

**Line 185:**

Similar to the previous comment for line 178. The authors suddenly report a spatial pattern.

We have removed the reference to the spatial pattern here.

**Line 189-193:**

Is a correlation alone enough to indicate agreement with the observations? A more comprehensive explanation would be helpful.

This part was significantly modified. We now compare with two observational products and provide a more detailed comparison:

> "The simulated $tCO_2$ uptake increases by only 0.003 $GtC/yr^2$ between 1980 and 1998 (Fig. 2e), in agreement with both observational estimates (Fig. 2f). While the simulated $tCO_2$ uptake decreases between 1998 and 2001 as in the observations, the magnitude of this simulated change is smaller than in the observational estimates.
>
> Similarly, while both simulation and observational estimates display an increase in $tCO_2$ uptake in the early 2000s, the reinvigoration only lasts until 2003 in the simulation, while it lasts until 2010 in both observational datasets. Finally, similar to the SOCAT only product, the simulation suggests a stagnation of the $tCO_2$ uptake between 2011 and 2018, while the SOCAT+SOCCOM product suggests a decrease in $tCO_2$ uptake.
>
> While the simulated $tCO_2$ changes are within the uncertainty range of the observational estimates (+/-0.15 GtC/yr) (Bushinsky et al., 2019) for most of the simulated period, the simulated variations are lower and outside of the uncertainty range between 1998 and 2005."

**Line 196:**

"The detrended tco2 ….. "

This statement needs to be linked to a particular figure for better understanding.

References to figures were added:

"The detrended $tCO_2$ flux (Fig. 2h) thus presents variations similar to $nCO_2$ (Fig. 2c)..."

**Line 205:**

"Changes in SST ….. "

This information is not visible or easily accessible to the reader.

This part of the manuscript was significantly changed and the figure now shows a map of the trend in pCO2 due to SST changes.

**Line 208:**

" On the other hand ….. "

The year 2015/16 is unexpectedly mentioned. What is the reason for this, and where can we locate this information?

This part of the manuscript was significantly changed and we are now discussing the overall $pCO_2$ trends between 1980 and 2021.

**Line 214:**

" The inter-basin….. "

This point requires a clearer association with the relevant data or figures.

This part of the manuscript was significantly changed, but the changes in westerly wind in the different basins are shown in Fig. S5 (now Fig. S6).

**Line 220-225:**

The explanation is not easily understandable.

This part of the manuscript was significantly modified.

**Line 250-251:**

"This is however ….. "

Why is this the case? Where can the reader locate information that supports this statement?

This part of the text was removed.

**Line 265:**

The abbreviation AABW has not been mentioned in the text previously.

AABW is now defined as Antarctic Bottom Water.

**Line 268:**

It's unclear from the sentence which figure supports this statement. What is the relation between oxygen and remineralized DIC in your model, can you provide some more explanation?

While remineralization of organic matter consumes oxygen, in this context we simply wanted to show that the changes in dissolved oxygen were also showing enhanced proportion of old waters within the SO upwelling branch, while the dissolved oxygen content was reduced within AABW and AAIW.

We are now only showing natural DIC and remineralized DIC anomalies. The text was amended accordingly (now in Section 3.3.1).

**Line 271-275:**

This paragraph cannot be fully understood or supported by the figures presented in the manuscript. If the authors claim a relationship between a specific variable and weak biological pump efficiency, they should provide a clear link between their statements and the relevant figures. However, it appears that the claim of weak biological pump efficiency is not presented or supported by any of the figures in the manuscript. Therefore, the authors should consider revising their analysis or adding a new figure to better support this claim or revise their statements to more accurately reflect the evidence presented in their figures. In general, it's important for authors to ensure that their claims are well-supported by the evidence they present and to provide clear links between their statements and the relevant figures to help readers understand and interpret their results.

As this section was significantly modified, this part of the text was deleted.

**Line 278:**

The abbreviation "NADW" is not introduced or defined in the manuscript.

The abbreviation NADW is removed from the manuscript, and North Atlantic Deep Water is spelled out.

**Line 281:**

"At both ….. "

What is visible? If it is visible, please provide a reference to the relevant figure to support your claim or statement.

A reference to Figure 8 is now added.

**Line 292-295:**

Please provide a clear link to the relevant figure to support your claim or statement.

Following a comment from Reviewer 2, we decided to remove that part of the manuscript.

**Line 296-299:**

How can the reader follow the CDW in Figure 7? It needs a better description of the figure.

Following a comment from Reviewer 2, we decided to remove that part of the manuscript.

**Discussion and conclusion:**

**Line 315-316:**

What are the numbers. Please quantify.

We have now also added numbers for the decadal-scale variability in SO tCO2 fluxes as inferred from observational estimates (0.25 GtC/yr, L. 360).

**Line 318-319:**

"It should be noted …. "

What are the authors trying to convey with it? It would be better to be more specific.

We have rephrased this sentence as follow:

" Such a mismatch between simulated SO tCO$_2$ variations and observations is prevalent in hindcast simulations (Gruber et al., 2019b), and could be due to an overestimation of the observed SO CO$_2$ flux variability (Gloege et al., 2021). The underestimation of the changes in tCO$_2$ uptake in the simulation could also be due a mis-representation of Southern Ocean stratification. "

**Line 320-321:**

"In addition, underestimation …. "

Does your model have this problem? If so, could you please mention it and show it first in the results section?

As indicated in the response to the comment on Line 189-193, this is now clearly mentioned in the Result section.

**Line 345:**

"we find that biological processes …. "

How did the authors reach this conclusion? There was not much related to biological processes throughout the results section. Could you please specify what you mean by 'biological processes'?

We are showing and discussing changes in detritus flux as a function of latitude for positive phases of the SAM compared to negative phases in figure 5. Now we are also showing the mean detritus flux as well as the changes occurring throughout the simulation. Changes in detritus flux are at least an order of magnitude lower than changes in air-sea CO2 fluxes as well as changes in DIC due to physical processes (i.e. Ekman pumping and diffusion at the base of the mixed layer). Nevertheless, this sentence was modified as follow:

As in previous studies, we find that changes in oceanic circulation are the primary driver of changes in SO CO$_2$ fluxes on decadal-time scales (Dufour et al., 2013, Resplandy et al., 2015, Nevison et al., 2019).

**Figures:**

Figure 1:

I suggest adding the PF and SAF contours to subfigure a.

This was added.

It would be helpful to provide the full names of the abbreviations 'PF' and 'SAF' in the figure label.

PF and SAF were re-defined in the figure caption.

Figure 2:

The x-axis should also include a tick mark for the year 1970.

The x-axis now includes a tick mark for the first year.

Figure 3:

The x-axis should also include a tick mark for the year 1980.

The x-axis now includes a tick mark for the first year.

Figure 4:

The unit of nCO2 flux (mol C/ m2/yr) like in Figure 1?

The units for the CO2 fluxes are now properly defined in all figure captions.

Figure 5:

What is the maximum and minimum extent of the y-axis in the plots?

What is meant by detritus flux, is it export production?

What is meant by "actual data"?

The maximum and minimum ticks were added to the y-axis. The caption was modified so that "actual data" is now replaced by "simulated fields". Within our modelling framework the detritus flux is similar to export production. This is now made clearer in the text.

Figure 7:

Why were these two time periods chosen for analysis?

Why is the contour in Figure 7(j) different from those in Figures 7(c), 7(f), and 7(l)?

What is it being compared to Gruber et al. 2019. ?

The manuscript was restructured and the analysis now focuses on the period 1980-2021. As such only the natural DIC anomalies are now shown in a figure that is being discussed with the processes leading to the long-term changes in natural CO2 flux. Since we are not showing

the total and anthropogenic DIC changes anymore, the comparison to Gruber et al., 2019 was removed.

Figure 8:

Please specify the time frame being discussed to avoid confusion.

The time period is now added in the caption of the figure.

---

## Author Comment (AC2)

Reviewer 2:

Menviel et al. analysed the Southern Ocean $CO_2$ sink using an eddy-rich global ocean biogeochemical model. Based on the results of their model, they argued that variations in the Southern Ocean $CO_2$ sink are mainly driven by changes in the outgassing of natural $CO_2$ and are related to the Southern Annular Mode (SAM). This variability in $CO_2$ flux could be explained by variations in surface dissolved inorganic carbon (DIC).

Such a modelling study, using a high-resolution ocean model, is essential as most of the currently used global ocean biogeochemical models cannot resolve eddies. The results presented in this study could help to improve our understanding of the Southern Ocean $CO_2$ sink. However, I have some questions about some of the results presented in this study (major comment). Therefore, the paper will probably be a significant scientific contribution with some revisions.

We thank the Reviewer for their helpful comments on our manuscript. We have made significant changes to the manuscript, including i) an improved and clearer comparison with observations, ii) an improved presentation of both full and detrended results, and iii) the inclusion of a similar simulation performed with the 1 degree version of the model.

We provide a point-by-point answer below in blue, with excerpts from the revised manuscript in green.

**Major comments:**

**1)** Authors mentioned that they model can reproduce some decadal variabilities of the Southern Ocean CO2 sink suggested by an observation-based product (i.e., SOM-FFN), and suggested an influence of the SAM, line 5: "*The simulated total CO2 flux exhibits decadal scale variability [...] in phase with observations and with variability in the Southern Annular Mode (SAM). Notably, a stagnation of the total CO2 uptake is simulated between 1982 and 2000, while a re-invigoration is simulated between 2000 and 2012.*"

These statements seem to be supported by the lines:

- Line 173: "*nCO2 fluxes are strongly correlated with the SAM index calculated from the JRA-55do dataset (R=0.62 for annual mean data and R=0.82 with a 5-year smoothing, Figs. 2b and S3)*"

- Line 192: "*The simulated and observational estimates of tCO2 flux are well correlated (R=0.55) and both display minimum tCO2 uptake in 2000-2001, and maximum in the early 1990s and early 2010s.*"

- Line 197: "*The nCO2 flux variability dominates the changes in tCO2 uptake with a strengthening of the winds and a poleward shift both reducing the tCO2 uptake (Figs. 2c,g and S3).*"

Did the authors remove the trends from the time series of nCO2, tCO2 (from their model and from SOM-FFN) and SAM before calculating the correlation coefficients? If not, the correlation coefficients between nCO2 and SAM, or between simulated and observed tCO2 estimates, are mainly influenced by the linear trend and do not provide information on the phasing between observed and simulated signals.

1) Regarding the $nCO_2$ fluxes. In the first version of the manuscript, we were only presenting non-detrended $nCO_2$ fluxes and SAM index. The reason behind this is that we are also interested in the multi-decadal-scale relation between the two. We however understand that the multi-decadal-scale increase in SAM could dominate the increase in SO $nCO_2$ outgassing. Therefore, to better highlight the short-term impact of the SAM, we are now showing the relationship between detrended $nCO_2$ fluxes and detrended SAM in the new Figure 3 (Figure R1) as well as scatter plots in Figure S4 (Figure R2).

Following some of the other comments from Reviewer 2 and as detailed below, we are also now focusing our analysis on the period 1980 to 2021.

The correlation between the detrended $nCO_2$ fluxes and SAM index is still significant at R=0.46, with increased $nCO_2$ outgassing during positive phases of the SAM. We also note that not only does the SAM index of the year impact $nCO_2$, but there is a "memory effect", with the SAM index of the previous year also modulating $nCO_2$. As such, if we plot the SO $nCO_2$ fluxes as a function of the detrended SAM index averaged over the current and previous year (as shown in Fig. R2, now Fig. S3), the correlation between the two is 0.8.

We have amended the text to accurately reflect the detrended and non-detrended relationships.

 "Since the SAM index displays a trend towards the positive phase between 1980 and 2021, the correlation mentioned above includes both interannual variability as well as decadal-scale changes. To also assess whether changes in the SAM significantly impact $nCO_2$ fluxes on an interannual timescale, we calculate the correlation between the detrended SAM index and detrended $nCO_2$ flux. The correlation is significant (p<0.05) and equals 0.46. We however note that if the detrended SAM index is averaged over two years (mean of the current and previous year), then the correlation equals 0.8 (Fig. S4a), indicating that the atmospheric forcing during the previous year also impacts surface natural $pCO_2$."

[Figure]

Figure R1: Detrended time-series of a) annual mean atmospheric $CO_2$ (ppm) used as forcing; b) SAM index calculated from the JRA55-do dataset (Stewart et al., 2020); Simulated integrated ocean to atmosphere $CO_2$ fluxes in the 0.1o (black) and 1o (blue) simulations: c) $nCO_2$, d) $aCO_2$, e) $tCO_2$. f) Detrended SO $tCO_2$ flux as derived from the SOM-FFN including both the SOCAT and SOCCOM data (red) (Bushinsky et al., 2019), and including the SOCAT data only (magenta) (Landschutzer et al., 2020). All the $CO_2$ fluxes are integrated over the SO (35oS-80oS) and are in GtC/yr. The correlation coefficients between the detrended SAM index and detrended $nCO_2$, $aCO_2$ and $tCO_2$ are 0.8, -0.42 and 0.69.

2) Regarding the $tCO_2$ fluxes, we are now showing and displaying correlation coefficients for both the detrended and non-detrended data.

The non-detrended SO $tCO_2$ fluxes have a correlation coefficient with the $tCO_2$ flux estimates of Bushinski et al., (2019) of 0.55, while the correlation coefficient with the estimates of Landschutzer et al., (2020) is 0.79.

We have added the following text to the manuscript:

"The simulated $tCO_2$ uptake increases by only 0.003 GtC/yr$^2$ between 1980 and 1998 (Fig. 2e), in agreement with both observational estimates (Fig. 2f). While the simulated $tCO_2$ uptake

decreases in between 1998 and 2001 as in the observations, the magnitude of this simulated change is smaller than in the observational estimates.

Similarly, while both simulation and observational estimates display an increase in $tCO_2$ uptake in the early 2000s, the reinvigoration only lasts until 2003 in the simulation, while it lasts until 2010 in both observational datasets. Finally, similar to the SOCAT only product, the simulation suggests a stagnation of the $tCO_2$ uptake between 2011 and 2018, while the SOCAT+SOCCOM product suggests a decrease in $tCO_2$ uptake.

While the simulated $tCO_2$ changes are within the uncertainty range of the observational estimates (+/-0.15 GtC/yr) (Bushinsky et al., 2019) for most of the simulated period, the simulated variations are lower and outside of the uncertainty range between 1998 and 2005."

The detrended data are less well correlated, at 0.35 for Bushinski et al (2019) and 0.37 for Landschutzer et al., (2020). We have added the following text:

"The correlation between detrended simulated and observationally estimated $tCO_2$ fluxes are 0.35 for SOCAT + SOCCOM (Bushinsky et al., 2019) and 0.37 for SOCAT only (Landschutzer et al., 2020). The two main disagreements between simulation and observations are in the mid 1990s and the late 2000s/early 2010s, when the ACCESS-OM2-01 simulates relatively low $tCO_2$ uptake (Fig.3e) while the observational estimates suggest high $tCO_2$ uptake (Fig. 3f). During these two periods the detrended $nCO_2$ fluxes are small, whereas the detrended $aCO_2$ fluxes are positive. These periods of low $tCO_2$ uptake in the model are thus due to reduced $aCO_2$ uptake, probably resulting from the atmospheric $CO_2$ forcing. "

The scatter plots of detrended $aCO_2$ and $tCO_2$ fluxes versus detrended SAM index are also shown in figure R2. The relationship between $aCO_2$ and SAM, with enhanced $aCO_2$ uptake during positive phases of the SAM is now significant at R=0.42. $nCO_2$ still dominates the $tCO_2$ relationship with the SAM, with a correlation coefficient of 0.69.

[Figure]

Figure R2: (from left to right) Detrended $nCO_2$, $aCO_2$ and $tCO_2$ fluxes versus detrended SAM index.

Furthermore, according to Figure 2, the stagnation in tCO2 uptake suggested by SOM-FFN is limited to the 1990s and not between 1982 and 2000 as the model simulated. In SOM-FFN, a reinvigoration occurred between 2000 and 2012, while the model simulated a reinvigoration only in the early 2000s (as the authors also mention in line 335: "*In agreement with observations, a re-invigoration of tCO2 uptake is simulated in the early 2000s.*"). Therefore, the statement "in phase with observations" in the abstract is misleading and does not seem to be supported by the authors' results. The relationships presented in this manuscript are

specific to their model and cannot be fully used to explain the variations in the Southern Ocean CO2 sink suggested by the observation-based method.

We have re-written that part of the Results as follow:

"The simulated $tCO_2$ uptake increases by only 0.003 $GtC/yr^2$ between 1980 and 1998 (Fig. 2e), in agreement with both observational estimates (Fig. 2f). While the simulated $tCO_2$ uptake decreases in between 1998 and 2001 as in the observations, the magnitude of this simulated change is smaller than in the observational estimates.

Similarly, while both simulation and observational estimates display an increase in $tCO_2$ uptake in the early 2000s, the reinvigoration only lasts until 2003 in the simulation, while it lasts until 2010 in both observational datasets. Finally, similar to the SOCAT only product, the simulation suggests a stagnation of the $tCO_2$ uptake between 2011 and 2018, while the SOCAT+SOCCOM product suggests a decrease in $tCO_2$ uptake.

While the simulated $tCO_2$ changes are within the uncertainty range of the observational estimates (+/-0.15 GtC/yr) (Bushinsky et al., 2019) for most of the simulated period, the simulated variations are lower and outside of the uncertainty range between 1998 and 2005."

If possible, and to better assess the added value of using a high-resolution ocean model, a comparison between tCO2 in the Southern Ocean simulated by the eddy-rich model presented here and by a global ocean biogeochemical model with lower spatial resolution should be added to Figure 2 (and in the manuscript).

Following from the Reviewer's suggestion, we are now including in Figures 2 (Figure R1) and the new Figure 3 (Figure R3) the results of a similar simulation performed with the 1 degree resolution version of the ACCESS-OM2. The 1 degree and 0.1 degree experiments are forced by the same JRA55-do forcing.

The time-evolution of the Southern Ocean $CO_2$ fluxes displays similar variability in both resolutions. While the 0.1 degree resolution provides a much better representation of small-scale processes and interaction with bathymetry thus providing a better representation of regional changes (Figs. 1 and 7), the 1 degree simulation captures well the large-scale processes (Figs. S4 and S8).

[Figure]

Figure R3: Non-detrended time series. Time series of a) annual mean atmospheric $CO_2$ concentration used as forcing, b) SAM index calculated from the JRA55-do dataset (Stewart et al., 2020). The horizontal dotted lines represent the thresholds used to define positive and negative SAM in the composites. Simulated integrated ocean to atmosphere $CO_2$ fluxes in the (annual mean in grey and 5-yr running mean in orange) 0.1° and (blue) 1° simulations: c) nCO2, d) aCO2, and e) tCO2. f) SO tCO2 flux as derived from the SOM-FFN (red) including both the SOCAT and SOCCOM data (Bushinsky et al., 2019), and (magenta) only including the SOCAT data (Landschutzer et al., 2020). The shading represents an uncertainty of 0.15 GtC/yr. All the CO2 fluxes are integrated over the SO (35°S-80°S) and are in GtC/yr. Dashed horizontal lines represent the 1980-2021 mean.

**2)** An important result from this modelling study is that "*The total SO CO2 uptake capability thus reduced since 1970 in response to a shift towards positive phases of the SAM.*" (line 13).

As mentioned by the authors in the introduction, Line 66: "*More recently, by analysing changes in SO tCO2 fluxes between 1980 and 2016, Keppler and Landschützer (2019) suggested that the net effect of the SAM on tCO2 uptake was nil and that instead the variability was arising from regional shifts in surface pressure linked to zonal wavenumber 3.*"

The authors need to discuss the discrepancy between their results and the results from Keppler and Landschützer (2019). Is a trend toward more positive SAM the only reason to explain a reduced CO2 uptake capability by the Southern Ocean since 1970? What about the other factors that could induce a long-term increase in the vertical stratification of the Southern Ocean and reduce its ability to absorb anthropogenic CO2 (e.g., Bourgeois T, Goris N, Schwinger J, Tjiputra JF. Stratification constrains future heat and carbon uptake in the Southern Ocean between 30°S and 55°S. Nat Commun. 2022, 13(1))? Although the SAM index could have an influence, it seems that other mechanisms can also influence the long-term changes in the Southern Ocean CO2 sink and need to be evaluated and discussed.

We agree with the Reviewer that changes in vertical stratification could impact $tCO_2$ uptake. We were already discussing this on L. 319-322, but we are now expanding the discussion by adding reference to Bourgeois et al., (2022) and more directly discussing the discrepancy with Keppler and Landschützer (2019).

"In addition, the underestimation of the simulated $tCO_2$ uptake in the late 2000s/early 2010s could be due a mis-representation of Southern Ocean stratification. It has indeed been suggested that the overturning rate of the lower cell was weaker during that time period (de Vries et al., 2017) due to enhanced stratification in the Southern Ocean (de Lavergne et al., 2014), linked to enhanced Antarctic basal melt rates (Adusumili et al., 2020). Enhanced stratification in the Southern Ocean would weaken the $aCO_2$ uptake (Bourgeois et al., 2022), but would reduce the $nCO_2$ outgassing (Menviel et al., 2015), thus potentially enhancing $tCO_2$ uptake."

and L. 370:

"This is in contrast to the conclusion of Keppler & Landschutzer, (2019) that the SAM had a net zero effect on SO $tCO_2$ uptake. Both our study and the one of Keppler & Landschutzer (2019) highlighted enhanced $tCO_2$ outgassing south of 50S during positive phases of the SAM as well as zonal asymmetries with a region of enhanced $tCO_2$ uptake in the Pacific sector of the SO. While Keppler & Landschutzer (2019) suggest this is linked to the zonal wave number 3 pattern, we attribute these asymmetries to the bathymetry and different poleward trends of the westerlies in the different sectors of the SO."

**Minor comments:**

**3)** Several references could be added in the introduction section and help the discussion. For example, studies that are partly based on observations and that have also demonstrated the influence of the SH westerlies on the air-sea CO2 flux:

- Gregor L, Kok S, Monteiro PMS. Interannual drivers of the seasonal cycle of CO2 in the Southern Ocean. Biogeosciences. 2018, 15(8), 2361–78.

- Nevison CD, Munro DR, Lovenduski NS, Keeling RF, Manizza M, Morgan EJ, et al. Southern Annular Mode Influence on Wintertime Ventilation of the Southern Ocean Detected in Atmospheric O2 and CO2 Measurements. Geophys Res Lett. 2020, 47(4), e2019GL085667.

An important modelling study that focuses on natural carbon variability:

- Resplandy L, Séférian R, Bopp L. Natural variability of CO2 and O2 fluxes: What can we learn from centuries-long climate models simulations? J Geophys Res Oceans. 2015, 120(1), 384–404.

The most recent review about the ocean CO2 sink variability:

- Gruber N, Bakker DCE, DeVries T, Gregor L, Hauck J, Landschützer P, et al. Trends and variability in the ocean carbon sink. Nat Rev Earth Environ. 2023, 4(2), 119–34.

We thank the referee for pointing us to these studies. We have now added some sentences in the Introduction to refer to the work of Gregor et al., (2018), Nevison et al., (2019), Resplandy et al., (2015) and Gruber et al., (2023).

**4)** Line 120: "Biogeochemical fields other than oxygen were initialised at the start of cycle 4 (1958). A uniform 0.01 mmol m−3 initial value was used for phytoplankton, zooplankton, detritus and CaCO3. […] Here, we skip the first twelve years of the fourth cycle (i.e. 1958-1970) from our analysis to allow the simulation to recover from the reset at the end of the previous cycle"

Twelve years is a relatively short period for the model to reach a steady state or recover from the reset. Could you provide in supplementary figures evidence that the biogeochemical fields have reach a steady state?

Could this influence the conclusion that (line 345) "we find that biological processes do not significantly impact air-sea CO2 fluxes on decadal-time scales, and that the changes in surface nDIC arise from changes in oceanic circulation"?

In the revised manuscript, we skip the first 22 years of the fourth cycle (i.e. 1958-1980) to allow the model to recover from the reset. This procedure follows the general protocol outlined by the phase II of the Coordinated Ocean-ice Reference Experiments.

We are also now including as figure S2 (Figure R4 here) the time evolution of nDIC, PO4 and O2 in the Southern Ocean and at different depth over the course of the experiment for both the 0.1 degree and 1 degree versions of the model.

The concentrations of the different tracers are not constant through the simulations since the atmospheric forcing varies (among other reasons). However, apart from surface PO4, the trends are much lower than 1%. The nDIC trends at the surface and in the deep are 0.02%, and 0.1% at intermediate depth. While the surface PO4 trend is 1.3% (which could also be due to the

trend towards a positive SAM), the trends at intermediate depth and at depth are of 0.1%. The O2 trend at the surface is 0.08%, while below 500m it is 0.8%.

As also seen in Figure R5, there is no significant trend in the Southern Ocean detritus concentration averaged between 40 and 100m depth (i.e. the location of the maximum detritus concentration).

As such, we think our models are equilibrated enough to assess the impact of recent changes in atmospheric forcing on Southern Ocean $CO_2$ fluxes.

[Figure]

Figure R4: Biogeochemical tracers time-series averaged over the Southern Ocean (35S-75S) in the ACCESS-OM2-01 (black) and ACCESS-OM2 (blue). (From left to right) nDIC, PO4 and O2 averaged over (top) the top 100m, (middle) between 500 and 1500m depth and (bottom) below 2000m depth.

[Figure]

Figure R5: Time-series of detritus concentration (mmol/m3) averaged over the Southern Ocean (35S-75S) and over 40-100m depth in the ACCESS-OM2-01 (black) and ACCESS-OM2 (blue).

**5)** Line 155: "…from autonomous biogeochemical floats (Gray et al., 2018; Prend et al., 2022)." In figure 1 caption, it says Bushinsky et al. (2019). Which one is used?

We are now being clearer and adding more information on the observational estimates that are used to compare with the model outputs. In Figure 1, we are now using version 2022 of Landschutzer et al., (2016 & 2020). In Figures 2 and 3, we are showing both Landschutzer et al., (2020) and Bushinsky et al., (2019).

**6)** Line 161: "…highlighting an uptake of aCO2 everywhere south of 35°S (Fig. 1d), with a maximum south of the PF (∼56.3∘S, Fig. S2d)." This is quite surprising. Normally, most of the aCO2 uptake should occur more north between the Polar Front and the Subpolar Front. For example, in:

- Gruber et al. (2019 – Annu. Rev. Mar. Sci.): "In contrast to natural CO2, the entire Southern Ocean south of 35°S is a sink for anthropogenic CO2 […] The majority of this uptake occurs between the Antarctic Polar Front and the Subpolar Front, leading to a distinct ring of high-uptake fluxes at the latitudes between 45°S and 55°S."
- See also figure 4 in Gruber et al. (2023 – Nat. Rev. Earth Environ.).

Could you explain the reason for this misrepresentation of the aCO2 uptake, and how is this impacting the conclusion (line 305) "the strengthening and poleward shift of the SH westerlies only had a small impact on aCO2 uptake "?

Gruber et al., (2023) indeed show an increase in aCO$_2$ uptake everywhere in the Southern Ocean since 1990, with a maximum at about 50S. This is in line with the simulation, even if in the simulation, there are two zonally-averaged maximum aCO$_2$ uptake at 42S and at 55S (old Fig. S2). It should be noted that the simulated and estimated changes in tCO$_2$ both suggest a maximum increase in tCO$_2$ uptake at about 40S. In the simulation, the aCO$_2$ changes are

obtained by subtracting the $nCO_2$ from the $tCO_2$. Similarly for observational products, assumptions have to be made to estimate the $aCO_2$ from the $tCO_2$.

The simulation suggests an increase in nCO2 outgassing south of 50S over the course of the simulation, with little changes in $tCO_2$. That indicates that there might also be an increase in $aCO_2$ uptake in that region. The increase in $nCO_2$ outgassing is linked to the enhanced upwelling, driven by the strengthening and poleward shift of the westerlies.

By comparing the detrended $aCO_2$ fluxes with the detrended SAM, we are now suggesting that positive phases of the SAM lead to enhanced $aCO_2$ uptake, even though the magnitude of that effect is still small (~25% of the $nCO_2$ change).

The text is modified to reflect this.

**7)** Line 176: "…similar correlation…" The correlation value needs to be provided in the text.

We removed that part of the text and instead mention in the methods that the SAM index calculated from the JRA-55do dataset captures well the SAM index based on observations (Marshall et al., 2003, Stewart et al., 2020).

**8)** Line 178: "The nCO2 outgassing occurs in…" and line 185: "The increase in aCO2 uptake occurs everywhere…" These sentences can be removed as the information was provided in the previous section 3.1.

These sentences were removed.

**9)** Line 183: "A weak correlation…" is the correlation statistically significant or not?

This was amended to:

"A weak but significant (p < 0.05) relationship…"

**10)** Figure 3 and Line 202: "As the outgassing of nCO2 occurs south of the SAF, we focus our analysis on that region. The natural pCO2 increase south of 50°S…". A clear definition and location of the front is provided (e.g., Figure 1). Instead of using the 50°S limit, the values should be averaged exactly is the area south of the front.

This figure as well as this section of the manuscript were significantly modified. The results are now shown as maps and not timeseries.

**11)** Section 3.4. "Changes in oceanic DIC". This section presents results which are not used in the following discussion. Furthermore, the figure 7 is the same as figure S8. These results need to be compared and discussed with published studies, otherwise this section should be removed.

The anthropogenic and total DIC shown in Figures 7 and S8 were different, as the mean trend was taken out from Fig. 7 whereas Fig. S8 was showing the full results. Nevertheless, the anthropogenic and total DIC are not shown anymore. Therefore Fig. S8 was removed and Fig. S7 was combined with Fig. 7. The results of this section are now moved earlier in the Results section.

---

## Author Response (AR2)

Dear Peter Landschützer,

Please find below the answer to the reviewer's comments, that have been fully implemented in the revised manuscript. We thank you and the reviewers for the time spent on our manuscript.

Best regards,
Laurie Menviel

Menviel et al. modified their manuscript following comments from two reviewers. Their responses are clear and they have made the requested changes. Comparisons between simulated air-sea CO2 fluxes and the SAM index and observation-based products are now more honest in highlighting the strengths and weaknesses of their simulations. Therefore, I believe that this paper will likely be a significant contribution and reach the quality standards for publication in the Biogeosciences journal after two technical corrections of the manuscript.

Minor comments:
1. The authors mentioned in their response that the "re-invigoration only lasts until 2003 in the simulation, while it lasts until 2010 in both observational datasets". But in the abstract, it says "while a revigoration is simulated between 2000 and 2012". Could the authors please correct the abstract?

The abstract has now been modified to: "Notably, two stagnations in tCO2 uptake are simulated: between 1982 and 2000, and between 2003 and 2011, while re-invigorations are simulated between 2000 and 2003, as well as since 2012."

2. The correlation between a two-year average SAM and nCO2 to suggest a "memory effect" is unclear. I'm not sure this is the right way to perform this analysis and support this claim. Instead, I would expect to get the results of a cross-correlation. If the results of a cross-correlation are not significant, I suggest removing this analysis from the manuscript. Furthermore, this claim is not discussed in the section "4 Discussion and conclusions".

This point was removed from the text. The figure S5 was modified so as to show the detrended fluxes versus the detrended SAM. The correlation coefficients were also amended in the main text.